# Riemannian Zeroth-Order Gradient Estimation with Structure-Preserving Metrics for Geodesically Incomplete Manifolds

**Shaocong Ma, Heng Huang**[*]
Department of Computer Science
University of Maryland
College Park, MD 20742, USA
{scma0908, heng}@umd.edu

## Abstract

In this paper, we study Riemannian zeroth-order optimization in settings where the underlying Riemannian metric $g$ is geodesically incomplete, and the goal is to approximate stationary points with respect to this incomplete metric. To address this challenge, we construct structure-preserving metrics that are geodesically complete while ensuring that every stationary point under the new metric remains stationary under the original one. Building on this foundation, we revisit the classical symmetric two-point zeroth-order estimator and analyze its mean-squared error from a purely intrinsic perspective, depending only on the manifold's geometry rather than any ambient embedding. Leveraging this intrinsic analysis, we establish convergence guarantees for stochastic gradient descent with this intrinsic estimator. Under additional suitable conditions, an $\epsilon$-stationary point under the constructed metric $g'$ also corresponds to an $\epsilon$-stationary point under the original metric $g$, thereby matching the best-known complexity in the geodesically complete setting. Empirical studies on synthetic problems confirm our theoretical findings, and experiments on a practical mesh optimization task demonstrate that our framework maintains stable convergence even in the absence of geodesic completeness.

## 1 Introduction

In this work, we consider the stochastic optimization problem on the smooth manifold $\mathcal{M}$ equipped with a Riemannian metric $g$:

$$\min_{p \in \mathcal{M}} f(p) = \mathbb{E}_{\xi \sim \Xi}[f(p; \xi)], \tag{1}$$

where $(\mathcal{M}, g)$ forms a $d$-dimensional Riemannian manifold, the individual loss $f(\cdot; \xi) : \mathcal{M} \to \mathbb{R}$ is a smooth function depending on a random data point $\xi$ drawn from a distribution $\Xi$. The Riemannian metric $g$ allows us for defining the first-order gradient $\nabla f(p; \xi)$ in the tangent space at each $p \in \mathcal{M}$, leading to the standard first-order Riemannian stochastic gradient method (Ring & Wirth, 2012; Bonnabel, 2013; Smith, 2014; Sato, 2021). In many practical scenarios, especially when the system incorporates non-differentiable external solvers or black-box objective functions especially when dealing with non-differentiable modules or black-box objective functions, the explicit gradient of the objective function is either unavailable or prohibitively expensive to compute. This practical challenge necessitates the use of zeroth-order optimization technique to approximate the gradient direction solely using the function evaluation (Nesterov & Spokoiny, 2017; Li et al., 2023b), given by

$$\widehat{\nabla} f(p; \xi) = \frac{f(\exp_p(\mu v); \xi) - f(\exp_p(-\mu v); \xi)}{2\mu} v, \tag{2}$$

where $v$ is a random vector sampled from a distribution over the tangent space $T_p \mathcal{M}$, and $\mu > 0$ is the perturbation stepsize. The exponential map $\exp_p : \mathcal{B} \subset T_p \mathcal{M} \to \mathcal{M}$ sends a tangent vector

---

[*]This work was partially supported by NSF IIS 2347592, 2348169, DBI 2405416, CCF 2348306, CNS 2347617, RISE 2536663.

$v \in T_p\mathcal{M}$ to the manifold $\mathcal{M}$ along the geodesic starting at $p$, with $\mathcal{B}$ denoting an open ball centered at the origin in $T_p\mathcal{M}$. In practice, the exponential map is often replaced by a first-order approximation known as a *retraction* (Definition B.3).

## 1.1 Challenges in Riemannian Zeroth-Order Optimization

While existing analyses of Riemannian zeroth-order optimization establish convergence guarantees under various algorithms and assumptions (Chattopadhyay et al., 2015; Fong & Tiňo, 2019; Wang et al., 2021a; Wang & Feng, 2022; Maass et al., 2022; Nguyen & Balasubramanian, 2023; Li et al., 2023b;a; Wang, 2023; He et al., 2024; Wang et al., 2023; Goyens et al., 2024; Zhou et al., 2025; Ochoa & Poveda, 2025), a fundamental yet often overlooked issue arises from the **local** nature of the exponential map (or, more generally, retractions). In practice, Riemannian zeroth-order methods often endow $\mathcal{M}$ with an *Euclidean* metric $g_E$ by viewing it as a submanifold of an ambient Euclidean space $\mathbb{R}^n$ and inheriting the metric from the embedding. This setting helps simplify numerical computations, but it has a **fundamental limitation**: the inherited Euclidean metric $g_E$ may not be *geodesically complete*. Specifically, for a point $p \in \mathcal{M}$, the exponential map $\exp_p$ is not necessary globally defined over the entire tangent space $T_p\mathcal{M}$. Consequently, a randomly sampled tangent vector $v \in T_p\mathcal{M}$ may fall outside the domain of $\exp_p$, making $\exp_p(v)$ **undefined**. Theoretically, one could instead begin with a *geodesically complete* metric, under which the exponential map $\exp : T\mathcal{M} \to \mathcal{M}$ is globally defined on the full tangent bundle $T\mathcal{M}$. The Nomizu-Ozeki theorem (Nomizu & Ozeki, 1961; Lee, 2018) guarantees the existence of such a complete metric on any smooth manifold without boundary. Then by applying the Nash embedding theorem (Nash, 1956), one could, in principle, obtain an equivalent geodesically complete Euclidean metric, allowing direct application of existing convergence analyses. **However**, the constructive proof of Nash's theorem is numerically nontrivial, making it infeasible for practical optimization algorithms.

This challenge motivates us to consider the following natural question:

> *Q:* How can we perform Riemannian zeroth-order optimization when the canonical Euclidean metric is geodesically incomplete?

To answer this question, we need to develop a Riemannian zeroth-order optimization algorithm for a given metric $g$ that may not be geodesically complete, yet remains capable of finding a stationary point. Our contributions are outlined in the following subsection.

## 1.2 Contributions

**Contribution 1 (Structure-Preserving Metric Construction):** To address the potential *geodesic incompleteness* of the given metric $g$, we construct the *structure-preserving metrics* $g'$ (Definition 2.5) in Theorem 2.6 that: *(i)* is geodesically complete, *(ii)* is conformally equivalent to the original metric $g$, and *(iii)* ensures any $\epsilon$-stationary point under $g$ is also an $\epsilon$-stationary point under $g'$. These properties allow us to work with the new metric $g'$ while maintaining the desired property as the original metric $g$. However, adopting the structure-preserving metric raises a fundamental challenge: the geometry induced by $g'$ generally differs from that of $g$. In particular, $g'$ is typically no longer an *Euclidean* metric inherited from the original ambient Euclidean space, which precludes the direct use of standard Riemannian zeroth-order gradient estimators (Li et al., 2023a;b). Overcoming this mismatch between estimator design and underlying geometry leads to our second contribution.

**Contribution 2 (Intrinsic Zeroth-Order Gradient Estimation):** Rather than finding a new ambient Euclidean space for the structure-preserving metric $g'$, we develop an *intrinsic* framework for zeroth-order optimization under non-Euclidean Riemannian metrics that relies solely on the manifold structure itself, and not on any embedding or representation in a larger ambient space. Under this intrinsic framework, we further analyze the mean-squared error (MSE) of the classical symmetric two-point zeroth-order gradient estimator (Equation (2)) under an arbitrary geodesically complete metric $g$ in Theorem 2.7, revealing the fundamental connection between the approximation error of gradient estimator and the curvature of the underlying manifold:

$$\mathbb{E}_{v \sim \text{Unif}(\mathbb{S}^{d-1})}\left[\left\|\widehat{\nabla}f(p;v) - \frac{1}{d}\nabla f(p)\right\|_p^2\right] \leq \frac{1+\mu^2\kappa^2}{d}\|\nabla f(p)\|_p^2 + \mathcal{O}(\mu^2).$$

where $v \sim \text{Unif}(\mathbb{S}^{d-1})$ is uniformly drawn from the unit sphere $\mathbb{S}^{d-1} \subset T_p\mathcal{M}$ induced by $g'$, $\widehat{\nabla}f(p;v)$ is the gradient estimator given by Equation (2), and $\kappa$ is a uniform upper bound on the

absolute sectional curvature of $(\mathcal{M}, g')$. In the flat case $\kappa = 0$, the bound reduces to the classical approximation error for zeroth-order gradient estimation in Euclidean spaces. Building on this result, Theorem 2.9 establishes the convergence of SGD under a general Riemannian metric $g$.

**Contribution 3 (Efficient Sampling under General Metrics):** Moreover, sampling uniformly from the unit sphere $\mathbb{S}^{d-1} \subset T_p\mathcal{M}$ with respect to a general Riemannian metric $g$ is nontrivial. We show that the commonly used rescaling approach (*i.e.* drawing a Gaussian vector and normalizing it to $g$-unit length) introduces an inherent bias under non-Euclidean metrics. To overcome this issue, we apply the rejection sampling method (Devroye, 2006) to Algorithm 1, an unbiased sampling procedure for generating $g$-unit-length tangent vectors. In Proposition 2.8, we prove that the output distribution of our method is exactly uniform over $\mathbb{S}^{d-1}$.

**Contribution 4 (Empirical Validation):** Lastly, to validate our theoretical results and demonstrate the empirical effectiveness of the proposed framework, we conduct extensive experiments on both synthetic and the practical experiments. Synthetic experiments examine: *(i)* the impact of sampling bias arising from rescaling sampling, and *(ii)* the influence of geometric curvature on estimation accuracy. In the mesh optimization task, our method further shows practical effectiveness in scenarios where geodesic completeness is absent.

## 1.3 APPLICATIONS

In this section, we highlight several applications of Riemannian zeroth-order optimization where the underlying manifold is geodesically incomplete.

**Mesh Optimization**  In physical simulations, mesh optimization is essential for improving discretized surface quality. Modern neural physical models, such as CFD-GCN (Belbute-Peres et al., 2020), adjust vertex positions by optimizing a quality metric, usually involving an external PDE solver. A major bottleneck is the requirement to implement auto-differentiation through this solver to obtain gradients, which is fundamentally difficult. Riemannian zeroth-order optimization offers a compelling alternative by avoiding this gradient calculation. In this setting, the manifold consists of the valid configuration space of vertex positions. This manifold, however, is geodesically incomplete under the Euclidean metric, because configurations on the boundary (e.g., a vertex on an edge) are excluded to prevent numerical instability.

**Irrigation System Layout Design**  This application seeks to optimize the physical coordinates of sprinklers to maximize water coverage. The coverage objective function is often a complex, non-differentiable simulation (e.g., modeling spray overlap, pressure, and wind), making it difficult to compute gradients. Riemannian zeroth-order optimization provides a gradient-free solution. The underlying manifold is the configuration space of valid sprinkler positions, defined by the open set within the field's boundaries. This manifold is geodesically incomplete, as typically we cannot directly put the sprinklers on the boundary of the field.

**Covariance Matrix Estimation**  This is a fundamental problem in multivariate statistics and machine learning, essential for tasks like PCA and Gaussian modeling. The goal is to find a matrix that best represents the data's covariance, often by minimizing a loss function (e.g., maximizing likelihood). The underlying manifold is the set of all $d \times d$ *positive definite matrices*, denoted $S_d^{++}$. A matrix $C$ is in this manifold if it is symmetric and $x^T C x > 0$ for all non-zero vectors $x \in \mathbb{R}^d$. This manifold is geodesically incomplete because it is an open convex cone.

In summary, the incompleteness in these examples poses a fundamental challenge, as existing literature typically requires geodesic completeness for gradient estimation. This limitation motivates our work to develop a framework that can perform Riemannian zeroth-order optimization without geodesic completeness.

## 2 MAIN RESULTS

In this section, we present the main results of this paper: *(i)* We propose the concept of structure-preserving metric (Definition 2.5) and provide its construction based on an arbitrary given metric $g$ (Theorem 2.6). *(ii)* Then we derive the approximation error upper bound of the two-point zeroth-order

gradient estimator *intrinsically*; that is, it does not rely on how the manifold is embedded into the ambient space (Theorem 2.7). *(iii)* To numerically obtain the gradient estimator under a general Riemannian metric $g$, we adopt the rejection sampling algorithm (Algorithm 1) to sample from the $g$-unit sphere. Later, Proposition 2.8 guarantees that the sampled vector satisfies the desired property. *(iv)* In Theorem 2.9, we establish the convergence of SGD under a general Riemannian metric $g$.

Here we summarize the assumptions used in our theoretical analysis. A brief manifold preliminary is included in Appendix B. Detailed discussions of each assumption are provided in Appendix C.1.

**Assumption 2.1.** *In the optimization problem given by Equation* (1)*, the individual loss function $f(\cdot; \xi) : \mathcal{M} \to \mathbb{R}$ satisfies the following two properties: (a) L-Bounded Hessian; for all $p \in \mathcal{M}$, the Hessian matrix at the point $p$ is bounded by $L$. (b) Lower boundedness; the infimum $f^*_\xi :=$ exists almost surely with $\xi \sim \Xi$.*

The following assumption imposes a regularization condition on the retraction used in Theorem 2.9. While it is always possible to construct a pathological retraction that deviates substantially from the exponential map, such choices may still scale with $\|v\|_p$ but would negatively affect the final convergence rate.

**Assumption 2.2.** *Let $f : \mathcal{M} \to \mathbb{R}$ be a smooth function. There exists a constant $C_{\mathrm{Ret}} \geq 0$ such that*

$$|f(\mathrm{Ret}_p(v)) - f(\exp_p(v))| \leq C_{\mathrm{Ret}}\|v\|_p^2.$$

**Assumption 2.3.** *There exist constants $\rho > 0$ and $M_3, M_4 > 0$ such that $\left\|\nabla^3 f(q)\right\|_{\mathrm{HS}} \leq M_3$ and $\left\|\nabla^4 f(q)\right\|_{\mathrm{HS}} \leq M_4$ for all $q \in \mathcal{B}_p(p, \rho)$, where $\mathcal{B}_p(p, \rho)$ denotes the geodesic ball of radius $\rho$ and $\|\cdot\|_{\mathrm{HS}}$ is the Hilbert-Schmidt norm.*

**Assumption 2.4.** *There exists a constant $\kappa \geq 0$ such that the sectional curvature of the Riemannian manifold $(\mathcal{M}, g)$ satisfies $|K_p(\sigma)| \leq \kappa$, for every point $p \in \mathcal{M}$ and every 2-plane $\sigma \subset T_p\mathcal{M}$. Equivalently, $-\kappa \leq K_p(\sigma) \leq \kappa$ for all $p$ and $\sigma$.*

## 2.1 STRUCTURE-PRESERVING METRIC

We begin with the definition of a *structure-preserving metric* associated with a given metric $g$. Since the exponential map of an arbitrary Riemannian metric $g$ is not necessarily globally defined on the entire tangent bundle $T\mathcal{M}$ (Proposition B.2), we seek an alternative metric $g'$ that is geodesically complete while preserving the essential geometric behavior of the original metric $g'$. This consideration motivates the following definition:

**Definition 2.5.** Let $(\mathcal{M}, g)$ be a Riemannian manifold. A Riemannian metric $g'$ is called *structure-preserving* with respect to $g$ if it satisfies:

(a) **(Geodesic completeness)** There exists $\rho > 0$ such that for any $p \in \mathcal{M}$, the domain of the exponential map $\exp_p : T_p\mathcal{M} \to \mathcal{M}$ contains the ball $\mathcal{B}_p(\rho) := \{v \in T_p\mathcal{M} : \|v\|_g \leq \rho\}$.

(b) **(Conformal equivalence)** There exists a positive smooth function $h : \mathcal{M} \to \mathbb{R}$ such that $g'_p(v, w) = h(p)g_p(v, w)$ for all $p \in \mathcal{M}$ and all $v, w \in T_p\mathcal{M}$.

(c) **($\epsilon$-stationarity preservation)** For any smooth function $f : \mathcal{M} \to \mathbb{R}$ and $\epsilon > 0$, every $\epsilon$-stationary point of $f$ under $g$ [1] is also an $\epsilon$-stationary point of $f$ under $g'$.

Here, we include a brief discussion on the motivation for introducing each condition: *(i)* The first condition (*geodesic completeness*) ensures that if we set the perturbation stepsize $\mu < \rho$ and fix the random vector $v$ on the $g$-unit sphere $\mathbb{S}^{d-1} \subset T_p\mathcal{M}$, the perturbed point $\mu v \in T_p\mathcal{M}$ will always be within the domain of the exponential map. *(ii)* The *conformal equivalence* condition preserves the set of stationary points; that is, for any smooth function $f : \mathcal{M} \to \mathbb{R}$, if $p$ is a stationary point under $g$, then it is also a stationary point under $g'$, and *vice versa*. *(iii)* The $\epsilon$-*stationarity preservation* condition gives rise to the name "stationary-preserving metric". It states that any $\epsilon$-stationary point under $g$ remains an $\epsilon$-stationary point under $g'$, ensuring that the transformation leaves the original set of $\epsilon$-stationary points unchanged. We emphasize, **however**, that the converse need not hold: an

---

[1]A point $p \in \mathcal{M}$ is called an $\epsilon$-*stationary point* of the smooth function $f$ under the Riemannian metric $g$ if the length of its gradient at $p$ is less than $\epsilon$; that is, $\sqrt{g_p(\nabla f(p), \nabla f(p))} < \epsilon$.

$\epsilon$-stationary point under $g'$ is generally not an $\epsilon$-stationary point under $g$. Nevertheless, under suitable conditions, this asymmetry does not affect the overall complexity guarantees as we will discuss it in Corollary 2.10.

In the following theorem, we demonstrate that given a metric $g$, it is always possible to construct a metric $g'$ which is *structure-preserving* with respect to $g$.

**Theorem 2.6.** *Let $\mathcal{M}$ be a smooth manifold (possibly non-compact), and let $g$ be any Riemannian metric on $\mathcal{M}$. Then there exists a Riemannian metric $g'$ on $\mathcal{M}$ which is structure-preserving with respect to $g$.*

*Proof.* The proof follows the classical construction presented by Nomizu & Ozeki (1961) with modifying the conformal coefficient $h : \mathcal{M} \to (0, +\infty)$ to ensure the $\epsilon$-stationarity preservation condition presented in Definition 2.5. The full proof is provided in Appendix C.3. $\square$

As illustrated in Figure 1b, the metrics constructed in this theorem ensure that geodesics remain within the manifold for all directions and lengths, eliminating concerns that random perturbations in zeroth-order gradient estimation could map outside the domain of the exponential map. Moreover, the conformal equivalence condition given by Definition 2.5 preserves the set of stationary points; therefore, in Riemannian zeroth-order optimization, it suffices to work with the new metric $g'$.

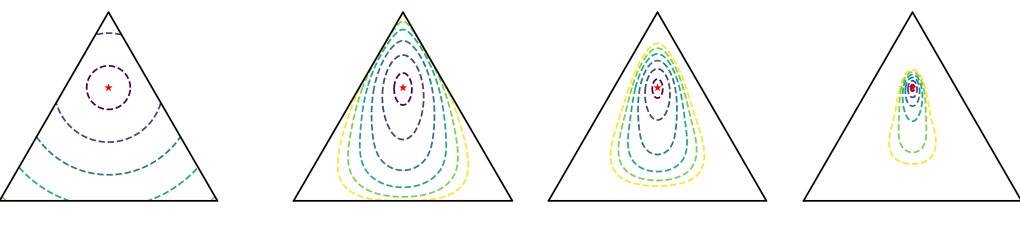

(a) Euclidean Metric  (b) Structure Preserving Metrics

Figure 1: Geodesic contours centered at $p = (0.2, 0.2, 0.6)$ under the Euclidean metric (Figure 1a) and three structure-preserving metrics (Figure 1b). Radii range from $0.1$ to $0.9$ in steps of $0.15$. Under each structure-preserving metric, geodesics from $p$ never exit the probability simplex, regardless of direction or length.

**Challenges Arising from the Structure-Preserving Metric**  Although Theorem 2.6 ensures that the constructed metric $g'$ satisfies the desired properties, existing results in Riemannian zeroth-order optimization cannot be applied directly to establish convergence guarantees under $g'$. This limitation arises because much of the current literature assumes a *Euclidean* setting, where $\mathcal{M}$ is embedded in a Euclidean space and the gradient estimation is determined by that embedding. In contrast, the new metric $g'$ is generally *non-Euclidean* with respect to the original ambient Euclidean space of $g$. To address this obstacle, we are motivated to develop an *intrinsic* zeroth-order optimization framework that operates solely on the manifold's geometry, without requiring $\mathcal{M}$ to be viewed as a subset of any Euclidean space.

### 2.2  Intrinsic Zeroth-Order Gradient Estimation under Non-Euclidean Metric

In this section, we introduce the intrinsic approach to estimate the gradient of the function $f : \mathcal{M} \to \mathbb{R}$ without relying on the ambient space. We take $g$ as a geodesically complete metric and consider the classical symmetric estimator

$$\widehat{\nabla} f(p) = \frac{f(\exp_p(\mu v)) - f(\exp_p(-\mu v))}{2\mu} v, \tag{3}$$

where $\exp_p : T_p\mathcal{M} \to \mathcal{M}$ is the exponential map. As noted by Bonnabel (2013), it is common to replace the exponential map with the retraction (Definition B.3).

The following theorem characterizes the mean-squared error (MSE) of this zeroth-order gradient estimator, establishing a connection between its approximation error and the intrinsic geometric

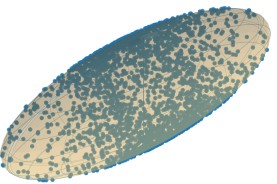
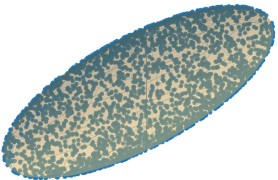

(a) Rescaling Sampling        (b) Rejection Sampling (Algorithm 1)

Figure 2: Illustration of sampling on the unit sphere induced by the non-Euclidean Riemannian metric $g$. The naïve rescaling sampler (Left Panel) produces a visibly non-uniform distribution, leading to a biased estimator. Our rejection sampler (Right Panel) presented in Algorithm 1 eliminates the bias and yields an even, truly uniform distribution.

properties of the underlying Riemannian manifold. The result is derived under the assumptions of bounded third- and fourth-order derivatives (Assumption 2.3) and globally bounded sectional curvature (Assumption 2.4). The full upper bound and the proof is deferred to Appendix C.4.

**Theorem 2.7.** *Let $(\mathcal{M}, g)$ be a complete $d$-dimensional Riemannian manifold and $p \in \mathcal{M}$. Let $f : \mathcal{M} \to \mathbb{R}$ be a smooth function and suppose that Assumptions 2.3 and 2.4 hold. Fix a perturbation stepsize $\mu > 0$ satisfying $\mu^2 \leq \min\{\frac{1}{d-1}, \frac{1}{2} + \frac{6}{d} + \frac{8}{d^2}\}$, and for any unit vector $v \in T_p\mathcal{M}$ define the symmetric zeroth-order estimator as in Equation (3). Then, for $v \sim \mathrm{Unif}(\mathbb{S}^{d-1})$ uniformly sampled from the $g_p$-unit sphere in $T_p\mathcal{M}$,*

$$\mathbb{E}_{v \sim \mathrm{Unif}(\mathbb{S}^{d-1})}\left[\left\|\widehat{\nabla} f(p; v) - \frac{1}{d}\nabla f(p)\right\|_p^2\right] \leq \frac{1 + \mu^2\kappa^2}{d}\left\|\nabla f(p)\right\|_p^2 + \mathcal{O}(\mu^2).$$

The bound in Theorem 2.7 reveals how the estimation error connects the intrinsic geometry of the manifold. In particular, the sectional curvature term $\kappa$ quantifies the influence of local geometry on the estimator's variance. When $\kappa = 0$, the curvature contribution disappears, and the bound reduces to the standard Euclidean variance expression.

### 2.3 SAMPLING FROM THE NON-EUCLIDEAN UNIT SPHERE

As the Riemannian metric $g$ defines a bilinear form on the tangent space $T_p\mathcal{M}$, uniformly sampling the $g$-unit sphere $\mathcal{B} := \{v \in T_p\mathcal{M} : g_p(v, v) = 1\}$ is equivalent to uniformly sample from the following compact set $\mathcal{C} := \{v \in \mathbb{R}^d : v^\top A v = 1\}$ for some positive definite matrix $A \in \mathbb{R}^{d \times d}$. The matrix $A \succ 0$ is determined by the Riemannian metric $g$ and the choice of local coordinates; in practice, we commonly use the local coordinate system spanned by the basis $\{\frac{\partial}{\partial x^i}|_p\}$. In this basis, the entries of $A$ are given by $A_{ij} := g_p(\frac{\partial}{\partial x^i}, \frac{\partial}{\partial x^j})$.

**Challenges in Sampling from the $g$-Unit Sphere** In Euclidean space, sampling from the unit sphere is relatively straightforward: one can sample from the standard Gaussian distribution and rescale the vector to have unit length. However, this method does not extend to the $g$-unit sphere. As illustrated in Figure 2, rescaling-based sampling results in points being predominantly concentrated along the minor axes. To achieve a truly uniform distribution over the $g$-unit sphere, we adopt the rejection sampling method in Algorithm 1 that generates random vectors uniformly distributed over the compact set $\mathcal{C}$.

The following proposition confirms that our sampling strategy yields the desired properties; that is, the resulting output $v$ exactly follows the uniform distribution over the unit sphere determined by the Riemannian metric $g$. The detailed proof is deferred to Appendix C.5.

**Proposition 2.8.** *Let the vector $v$ be generated by Algorithm 1. Then it follows the uniform distribution over the compact set $\mathcal{C} := \{v \in \mathbb{R}^d : v^\top A v = 1\}$.*

---

**Algorithm 1:** Uniform Sampling on Ellipsoid $\mathcal{C} = \{x \in \mathbb{R}^d \mid x^\top A\, x = 1\}$

---

**Input:** A positive definite matrix $A \in \mathbb{R}^{d \times d}$
**Output:** $v \in \mathbb{R}^d$

1   $Q, \Lambda \leftarrow \mathrm{eig}(A)$     `// Eigenvalue Decomposition:`   $A = Q\Lambda Q^\top,\ \Lambda = \mathrm{diag}(\lambda_1, \ldots, \lambda_d)$
2   $L \leftarrow Q\, \Lambda^{-1/2},\ \lambda_{\max} = \max\{\mathrm{diag}(\Lambda)\}$
3   **while** `True` **do**
4      Draw $z \sim \mathcal{N}(0, I_d)$; set $s \leftarrow z/\|z\|$; $v \leftarrow L\, s$         `// proposal point on` $\mathcal{C}$
5      Draw $u \sim \mathcal{U}(0, 1)$
6      **if** $u < \sqrt{v^\top A^2 v / \lambda_{\max}}$ **then**
7         return $v$

---

## 2.4   Convergence of Zeroth-Order SGD under Non-Euclidean Metric

In the previous section, we have shown that the accuracy of Riemannian zeroth-order gradient estimator is improved as the underlying geometric structure selected to be flatter. However, there is no free lunch in simply flattening the manifold. As we have commented in Definition 2.5, the $\epsilon$-stationary point under the new metric $g'$ may not be the $\epsilon$-stationary point under the original metric $g$; so one must balance estimator accuracy with optimization dynamics.

To solve the optimization problem in Equation (1), we employ the SGD algorithm. Starting from an initial parameter $p_1$, the updates are given by

$$p_{t+1} = \mathrm{Ret}_{p_t}\!\big(\eta\, \widehat{\nabla} f(p_t; \xi_t)\big), \tag{4}$$

for $t = 1, 2, \ldots, T-1$, where $\mathrm{Ret} : T\mathcal{M} \to \mathcal{M}$ is the retraction (Definition B.3), $\eta \in \mathbb{R}$ is the learning rate, $\{\xi_t\}_{t=1}^T$ is the stochastic data sample accessed at the $t$-th update, and $\widehat{\nabla} f(p_t; \xi_t)$ is the Riemannian zeroth-order gradient estimation of $f(\cdot; \xi_t)$ at the point $p_t$ defined as

$$\widehat{\nabla} f(p) = \frac{f(\mathrm{Ret}_p(\mu v)) - f(\mathrm{Ret}_p(-\mu v))}{2\mu}\, v, \tag{5}$$

Now we build the convergence analysis of Riemannian SGD algorithm. We write $a \lesssim b$ if there exists a constant $\mathsf{C} > 0$ such that $a \leq \mathsf{C}\, b$. The constant $\mathsf{C}$ may depend only on fixed problem parameters. Besides the boundedness assumption made in Theorem 2.7, we additionally require the $L$-smoothness (Assumption 2.1) and the regularization condition on the retraction (Assumption 2.2).

**Theorem 2.9.** *Let $(\mathcal{M}, g)$ be a geodesically complete $d$-dimensional Riemannian manifold. Let $f : \mathcal{M} \to \mathbb{R}$ be a smooth function and suppose that Assumptions 2.1 to 2.4 hold. Define the symmetric zeroth-order estimator as in Equation (5). Let $\{p_t\}_{t=1}^T$ be the SGD dynamic finding the stationary point of Equation (1) generated by the update rule Equation (4) with requiring $\eta \lesssim \sqrt{\frac{d}{T}}$ and $\mu^2 \lesssim \sqrt{\frac{d}{T}}$ (specified in Equation (21)), then there exists constants $\mathsf{C}_1, \mathsf{C}_2, \mathsf{C}_3 > 0$ such that*

$$\min_{1 \leq t \leq T} \|\nabla f(p_t)\|_{p_t}^2 \leq \mathsf{C}_1\, \frac{d}{\eta T} + \mathsf{C}_2\, \eta + \mathsf{C}_3\, d^2 \mu^2.$$

*In particular, choosing $\mu \lesssim \frac{1}{d^2}\sqrt{\frac{d}{T}}$ yields $\min_{1 \leq t \leq T} \|\nabla f(p_t)\|_{p_t}^2 \lesssim \sqrt{\frac{d}{T}}$.*

*Proof.* The proof directly follows the standard convergence analysis of SGD in Euclidean space (Mishchenko et al., 2020). We may further relax the $L$-smoothness assumption to the expected smoothness condition proposed by (Khaled & Richtárik, 2022). The zeroth-order gradient approximation error term is bounded using Theorem 2.7. See Appendix C.6 for the full proof.    □

**Importantly**, the upper bound in Theorem 2.9 is not our final goal. We typically begin with a canonical Euclidean metric $g_E$, which may fail to be geodesically complete. To overcome this issue, we construct a new metric $g := h g_E$ via Theorem 2.6 and then apply the convergence analysis under this new metric $g$ (using Theorem 2.7 and Theorem 2.9). However, an $\epsilon$-stationary point delivered by

SGD under $g$ often is not an $\epsilon$-stationary point under $g_E$, unless the additional condition stated in the following corollary is imposed:

**Corollary 2.10.** *Let $g_E$ be the Euclidean metric, and let $g$ be a structure-preserving metric with respect to $g_E$. Under the same assumptions as Theorem 2.9, suppose that **either** of the following conditions holds:*

*(a) $g_E$ is geodesically complete; or*

*(b) the set of $\epsilon$-stationary points under $g_E$, $K := \{\, p \in \mathcal{M} : \|\nabla_{g_E} f(p)\|_{p,g_E} \leq \epsilon \,\}$, is compact.*

*Then it requires at most $T \leq \mathcal{O}\left(\frac{d}{\epsilon^4}\right)$ iterations to achieve $\min_{1 \leq t \leq T} \mathbb{E}\left[\|\nabla f(p_t)\|_{p_t,g_E}^2\right] \leq \epsilon^2$.*

*Proof.* Under either condition, the conformal coefficient $h$ constructed in Theorem 2.6 admits a uniform upper bound. Consequently, an $\epsilon$-stationary point with respect to the new metric $g := h g_E$ is also an $\epsilon$-stationary point with respect to the original metric $g_E$, up to a constant scaling factor. This structure allows the complexity bound established in Theorem 2.9 to transfer directly to the metric $g_E$. See Appendix C.7 for the full proof. □

Item (a) corresponds to the classical setting in which the original metric is geodesically complete. Item (b), on the other hand, specifies conditions under which an $\epsilon$-stationary point under the new metric is also an $\epsilon$-stationary point under the original metric. We emphasize that Theorem 2.9 establishes convergence even in more general scenarios, though with potentially worse complexity bounds than in the geodesically complete case. This phenomenon highlights a key distinction between the framework studied in our work and the traditional geodesically complete setting. Building on this result, we extend the best-known complexity bound for Riemannian zeroth-order SGD on smooth objectives from the special case of manifolds equipped with a Euclidean metric to a much broader class of manifolds endowed with general Riemannian metrics.

## 3 EXPERIMENTS

In the experimental section, we aim to validate the theoretical findings presented in Section 2. The two synthetic experiments are designed to investigate the following questions: *(i)* How does sampling bias influence the convergence behavior of Riemannian zeroth-order SGD? *(ii)* How does the curvature of the underlying manifold affect the accuracy of gradient estimation? In addition, we conduct a real-world experiment on mesh optimization (Hoppe et al., 1993; Belbute-Peres et al., 2020; Ma et al., 2025), a practical application in which the positions of nodes are naturally represented as points on the probability simplex.

### 3.1 SYNTHETIC EXPERIMENT: IMPACT OF SAMPLING BIAS

In this experiment, we investigate the impact of sampling bias in zeroth-order Riemannian optimization. Specifically, we consider two objective functions defined on the Euclidean space $\mathbb{R}^d$, equipped with a non-Euclidean Riemannian metric given by $g_A(u,v) := u^\top A v$:

$$f_{\text{quadratic}}(x) = \frac{1}{2}\mathbb{E}_\xi\, x^\top(B+\xi)x, \qquad f_{\text{logistic}}(x) = \mathbb{E}_{(\zeta,y)}\, \log(1 + \exp(-y\,\zeta^\top x)) + \frac{\lambda}{2}x^\top Bx,$$

where each entry of $\xi$ is independently drawn from $\mathcal{N}(0,1)$, and $(\zeta,y)$ is sampled from a fixed categorical data distribution. The matrix $B \in \mathbb{R}^{d \times d}$ is a pre-generated positive definite matrix. We compare two sampling strategies for Riemannian gradient estimation in the zeroth-order setting: *(i) Rejection sampling* (Algorithm 1), which produces uniform samples from the Riemannian unit sphere and is unbiased as shown in Proposition 2.8. *(ii) Rescaling sampling*, which samples a Gaussian vector then normalizes it to the unit sphere with respect to the Riemannian metric $g_A$.

**Experimental Implications** For each configuration, we report the average objective value over 16 independent runs using the same hyperparameter settings for the SGD optimizer. As shown in Figure 3, the rejection sampling method (Algorithm 1) consistently outperforms the traditional rescaling approach; the rescaling method even leads to divergence under the same hyper-parameter

setting for the logistic loss objective (right panel of Figure 3). These results highlight the importance of using Algorithm 1 to ensure an unbiased uniform distribution over the Riemannian $g$-unit sphere, which is critical for stable and effective training. The complete experimental details are included in Appendix D.1.

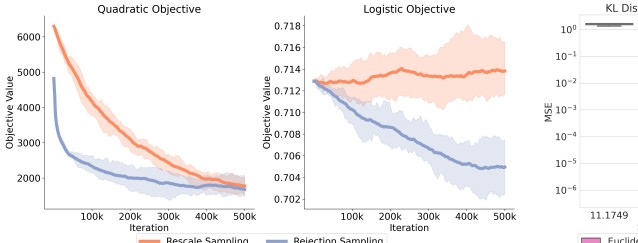
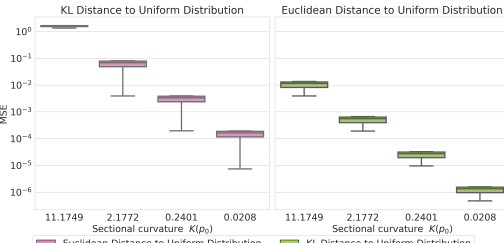

Figure 3: The impact of sampling bias on the convergence of Riemannian zeroth-order SGD.

Figure 4: The impact of sectional curvatures on the gradient estimation accuracy.

## 3.2 SYNTHETIC EXPERIMENT: IMPACT OF SECTIONAL CURVATURE

In this experiment, we investigate the impact of sectional curvature on the accuracy of zeroth-order gradient estimation. Specifically, we evaluate gradient estimation errors at a fixed point $p_0$ under four conformally equivalent Riemannian metrics with different curvatures. We consider two objective functions commonly used in the optimization problem on probability simplex:

$$f_{\text{KL}}(p) = \text{KL}(p\|q) = \sum_i p_i \log\left(dp_i\right), \qquad f_{\text{Euclidean}}(p) = \frac{1}{2}\|p - q\|^2 = \frac{1}{2}\sum_{i=1}^{d}(p_i - \frac{1}{d})^2$$

where $q = \frac{1}{d}\mathbf{1}_d$ denotes the centroid of the simplex. We measure the accuracy of gradient estimation using the mean-squared error (MSE) under its own Riemannian metric, computed over 50,000 independent trials of zeroth-order gradient estimation (Equation (3)). The complete experimental details are included in Appendix D.2.

**Experimental Implications** As depicted in Figure 4, the Riemannian MSE of zeroth-order gradient estimation decreases as the sectional curvature $K(p_0)$ decreases. This empirical finding aligns with our theoretical upper bound presented in Theorem 2.7, illustrating a clear connection between gradient estimation accuracy and the intrinsic geometric properties of the underlying manifold. In particular, higher curvature consistently results in larger estimation errors for both objective functions.

## 3.3 GRADIENT-BASED MESH OPTIMIZATION

In modern physical simulation, solving PDEs often relies on finite-volume methods with spatial discretizations and external solvers that lack automatic differentiation support (Belbute-Peres et al., 2020; Ma et al., 2025), making the zeroth-order approach an ideal tool for optimizing mesh positions.

In this experiment, we consider the gradient-based mesh optimization problem for solving the Helmholtz equation (Goodman, 2017; Engquist & Zhao, 2018),

$$\nabla^2 f = -k^2 f,$$

where $\nabla^2$ denotes the Laplace operator, $k = 10$ is the wave number, and $f$ is the eigenfunction. The ground-truth solution is computed on a fine mesh with resolution $200 \times 200$. Our goal is to optimize the node positions of a regular coarse mesh with resolution $20 \times 20$ so that its performance approximates that of the ground-truth solution. The mesh node (in our setting, boundary nodes are fixed and excluded from optimization) is represented using a simplex formulation: each trainable node $p = (x, y)$ is expressed as a convex combination of its six neighbors under the regular triangular initialization. This parameterization naturally leads to a manifold optimization problem. **However**, the coordinate simplex, under its canonical embedding, is geodesically incomplete. To ensure the exponential map remains well-defined and to prevent perturbed nodes from crossing mesh edges, we

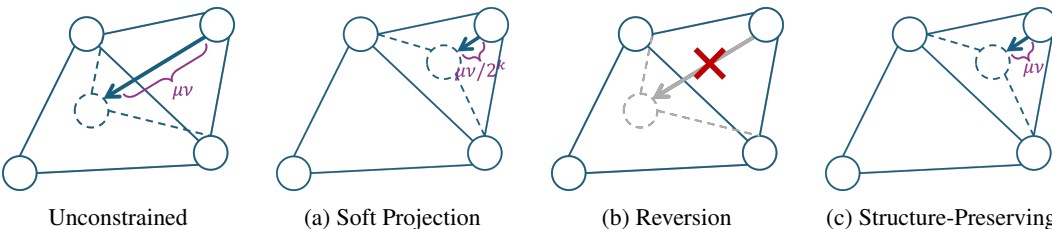

| Unconstrained | (a) Soft Projection | (b) Reversion | (c) Structure-Preserving |

Figure 5: The leftmost panel illustrates an invalid optimization step on a mesh node; it crosses the edge, causing potential error in the external PDE solver. Figure (a) illustrates the *Soft Projection* approach, which resolves the issue by repeatedly reducing the perturbation stepsize $\mu$ along the perturbation direction $v$ until the movement becomes valid. Figure (b) shows the *Reversion* approach, which instead handles invalid steps by reverting to the original position. Figure (c) takes the advantage of the structure-preserving metric, which twists the underlying Riemannian structure ensuring that the perturbation won't move the point out of the domain.

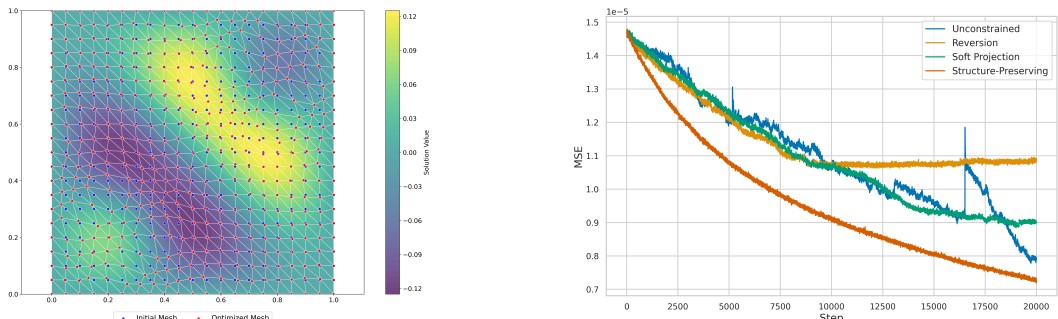

Figure 6: The left panel shows the ground-truth prediction (background), the initial mesh (blue), and the optimized mesh (red) using our proposed method. The nodes adaptively concentrate around the critical region while preserving the overall mesh structure. The right panel presents the loss curves for different approaches. Our method achieves both stable and efficient convergence.

adopt our proposed structure-preserving approach and compare it against several natural baselines, as illustrated in Figure 5.

Figure 6 presents the loss curves of the up-sampled prediction over 20,000 optimization steps. The *unconstrained* method often violates mesh validity, leading to unstable fluctuations, most notably around the 16,000th step. The *reversion* prevents invalid updates but quickly stalls after 8,000 steps; similarly, the *soft projection* stabilizes training but progresses slowly, showing little improvement beyond 14,000 steps. In contrast, our *structure-preserving* approach consistently reduces the error throughout training, achieving the lowest final MSE without instability. These findings highlight that structure-preserving approaches not only maintain feasibility but also enable effective convergence.

## 4 CONCLUSION

In this work, we consider the zeroth-order optimization problem on Riemannian manifolds when the underlying metric might be **geodesically incomplete**. We propose the structure-preserving metric that is geodesically complete, while preserving the original set of stationary points (Theorem 2.6). Building on this foundation, we intrinsically derive the accuracy upper bound of the classical two-point gradient estimator and reveal the role of manifold curvature (Theorem 2.7). We further propose an unbiased rejection sampling scheme for generating perturbation directions under general Riemannian metrics (Proposition 2.8). Our theoretical analysis establishes convergence guarantees that extend the best-known complexity results beyond the Euclidean setting to a broader class of Riemannian manifolds (Theorem 2.9). Empirical studies, including synthetic experiments and a mesh optimization task, demonstrate that structure-preserving approaches enable stable and effective convergence. These findings extend the theoretical understanding of zeroth-order optimization methods in Riemannian manifolds and provide practical tools for Riemannian black-box optimization.

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

# Appendix

## Table of Contents

## A  RELATED LITERATURE

### A.1  OPTIMIZATION ON RIEMANNIAN MANIFOLDS

**First-Order Methods**  Riemannian first-order optimization adapts gradient-based methods to Riemannian manifolds. For geodesically convex functions, Riemannian gradient descent enjoys convergence guarantees akin to Euclidean GD, with complexity $O(L/\epsilon)$ for $L$-smooth objectives. Zhang & Sra (2016) established global complexity bounds on Hadamard manifolds with curvature-dependent rates. Stochastic Riemannian gradient descent converges almost surely under standard assumptions (Bonnabel, 2013), while variance-reduced variants such as R-SVRG (Zhang et al., 2016) and R-SRG/SPIDER improve convergence for finite-sum problems. Adapting acceleration (Nesterov, 2013a; 1983; 2013b) to manifolds proved challenging due to the absence of global linearity. Early methods (Liu et al., 2017) were shown computationally impractical; Zhang & Sra (2018) and Ahn & Sra (2020) addressed this issue by controlling metric distortion, achieving accelerated rates under bounded curvature. Alimisis et al. (2021) proposed momentum-based RAGDsDR, while Kim & Yang (2022) achieved optimal accelerated rates with RNAG, matching the $O(\sqrt{L/\epsilon})$ Euclidean complexity. There are still some fundamental limits remained: Hamilton & Moitra (2021) and Criscitiello & Boumal (2022) showed that curvature may prevent acceleration entirely on negatively curved manifolds. These negative impacts would be eliminated using the second-order methods.

**Second-Order Methods**  Riemannian second-order methods utilize curvature via Hessians and connections. Newton-type methods achieve quadratic local convergence using the Riemannian

Hessian (Absil et al., 2008), though global convergence requires safeguards like line search or trust-region strategies. Trust-region methods (Absil et al., 2007) solve quadratic models in the tangent space and retract back, ensuring convergence to second-order points. Recent improvements analyze their behavior near strict saddles (Goyens & Royer, 2024). Alternatively, Riemannian ARC (Agarwal et al., 2021) uses cubic regularization to achieve optimal $O(\epsilon^{-3/2})$ complexity. Quasi-Newton methods generalize BFGS to manifolds via vector transports. Ring & Wirth (2012) initiated this line, and Huang et al. (2018) showed global convergence (and superlinear rates) under mild assumptions. Limited-memory variants (R-LBFGS) scale better to large problems. Overall, second-order methods offer faster local convergence but require careful geometric handling of Hessians and transports.

**Zeroth-Order Methods**    When gradients are unavailable, zeroth-order methods estimate descent directions via sampling. Li et al. (2023b;a) applied Gaussian smoothing in tangent spaces using exponential maps to construct unbiased gradient estimators with variance bounds that depend on curvature and dimension. A stochastic zeroth-order Riemannian gradient descent achieves $O(n/\epsilon^2)$ convergence for smooth nonconvex functions. Wang et al. (2023) proposed two-point bandit methods (R-2-BAN) for online geodesically convex optimization, showing regret bounds matching Euclidean rates up to curvature factors. Other derivative-free approaches include retraction-based direct search methods, as in Kungurtsev et al. (2023), with convergence guarantees for smooth and nonsmooth objectives. Yao et al. (2021) developed a Polak–Ribiére–Polyak conjugate gradient method using only function values and nonmonotone line search, achieving global convergence and hybridizing with Newton steps for improved performance.

**Hybrid and Other Emerging Directions**    Several novel methods extend optimization frameworks to the Riemannian setting. Adaptive methods such as Riemannian Adagrad and Adam (Bécigneul & Ganea, 2019) address the challenge of accumulating gradients across varying tangent spaces by working on product manifolds, yielding convergence results for geodesically convex problems. Riemannian conjugate gradient (CG) methods, which define conjugacy across tangent spaces via vector transport, have been shown to converge globally under standard line-search assumptions (Sato & Iwai, 2013; Sato, 2022; Kim & Yang, 2022). Projection-free methods like Riemannian Frank-Wolfe avoid expensive retractions by solving a linear oracle at each step. Weber & Sra (2023) showed that Riemannian Frank-Wolfe method converges sublinearly in general and linearly under geodesic strong convexity. For composite objectives with nonsmooth regularizers, Riemannian proximal gradient methods offer convergence guarantees; Huang & Wei (2022) proved an $O(1/k)$ rate under retraction-based convexity. Finally, primal-dual interior-point methods have also been adapted: Lai & Yoshise (2024) introduced a Riemannian interior-point algorithm with local superlinear convergence and global guarantees, mirroring the classical barrier method behavior in curved spaces.

## A.2    RIEMANNIAN ZEROTH-ORDER GRADIENT ESTIMATORS

In this section, we discuss several widely used gradient estimators in Riemannian optimization and highlight their connections to our work. Importantly, all of these estimators are developed under the assumption of a complete Riemannian manifold. In contrast, our setting differs from this convention by considering optimization over possibly *geodesically incomplete* Riemannian manifolds.

**Wang et al. (2021b)**    This paper extends the *one-point* bandit estimator to homogeneous Hadamard manifolds. At the point $x \in \mathcal{M}$ and given $y$ uniformly sampled from the geodesic sphere centered at $x$ with the radius $\delta$, by using the gradient estimator

$$\widehat{\nabla} f(x) := f(y) \frac{\exp_x^{-1}(y)}{\| \exp_x^{-1}(y) \|},$$

this work established the best-possible regret rate $\mathcal{O}(T^{3/4})$ for $g$-convex losses in the online regret optimization problem.

**Wang et al. (2023)**    This journal version further develops a *two-point* bandit estimator on symmetric Hadamard manifolds. Uniformly draw $y$ from the geodesic sphere centered at $x$ with the radius $\delta$ and defined $-y$ as the antipodal point of $y$. The gradient estimator is given by

$$\widehat{\nabla} f(x) := \frac{f(y) - f(-y)}{2} \frac{\exp_x^{-1}(y)}{\| \exp_x^{-1}(y) \|}.$$

The regret improves to $\mathcal{O}(\sqrt{T})$ for $g$-convex and $\mathcal{O}(\log T)$ for strongly $g$-convex losses.

**Li et al. (2023b) & Maass et al. (2022)**  These papers introduce a non-symmetric two-point Riemannian zeroth-order oracle for the online setting (Maass et al., 2022) and the expected loss setting (Li et al., 2023b). With a tangent perturbation $v \in T_x\mathcal{M}$ (obtained by projecting an ambient Gaussian onto $T_x\mathcal{M}$), the gradient estimator is

$$\widehat{\nabla} f(x) := \frac{f \circ \exp_x(\mu v) - f(x)}{\mu} v.$$

Here we have adjusted the estimator from Maass et al. (2022) to the time-invariant expected objective function setting to align with our problem setup. This estimator is the direct generalization of the one-side Gaussian smoothing estimator widely used in Euclidean zeroth-order optimization.

**He et al. (2024)**  This work extends coordinate-wise finite differences to manifolds. Using an orthonormal basis $\{e_i\}$ of $T_x\mathcal{M}$,, the deterministic coordinate-wise zeroth-order estimator is

$$\widehat{\nabla} f(x) := \sum_{i=1}^{d} \frac{f \circ \exp_x(\mu e_i) - f \circ \exp_x(-\mu e_i)}{2\mu} e_i.$$

In summary, compared to approaches that rely on projecting from the ambient Euclidean space, our analysis is purely *intrinsic*, that is, the gradient estimator depends only on the Riemannian structure and is independent of any particular embedding. In contrast to prior intrinsic estimators, which primarily focus on geodesically convex problems, our work addresses the non-convex setting. As a result, our contributions extend the scope of existing research on Riemannian zeroth-order optimization.

## B  PRELIMINARIES

In this section, we review some basic definitions and results from Riemannian geometry that are used in our analysis. For a full review, we refer the reader to some classical textbook (Lee, 2003; 2018). For convenience, we summarize our notations in Table 2.

**Smooth Manifolds**  A $d$-dimensional *smooth manifold* $\mathcal{M}$ is a second-countable Hausdorff topological space such that at any point $p \in \mathcal{M}$, there exists $U_p \subset \mathcal{M}$, a neighborhood of $p$, such that $U_p$ is diffeomorphism to the Euclidean space $\mathbb{R}^d$. Let $C^\infty(U)$ be all smooth functions over $U \subset \mathcal{M}$. A *deviation* at $p \in \mathcal{M}$ is a linear mapping $v : C^\infty(U_p) \to \mathbb{R}$ satisfying

$$v(fg) = v(f) \cdot g(p) + v(g) \cdot f(p)$$

for all $f, g \in C^\infty(U_p)$. Then the *tangent space* at $p$, denoted by $T_p\mathcal{M}$, is the real vector space of all deviation at $p$. The *tangent bundle* is the disjoint union of all tangent spaces

$$T\mathcal{M} := \{(p,v) \mid p \in \mathcal{M}, v \in T_p\mathcal{M}\}.$$

A smooth map $f : \mathcal{M} \to \mathbb{R}^n$ is called an *immersion* if its differential $df|_p : T_p\mathcal{M} \to T_{f(p)}\mathbb{R}^n$, defined by $df|_p(v) := v(f)$ for each $v \in T_p\mathcal{M}$, is an injective function at every $p \in \mathcal{M}$; it is called an *embedding* if it is an immersion and is also homeomorphic onto its image $f(\mathcal{M}) := \{f(p) \mid p \in \mathcal{M}\}$.

**Riemannian Manifolds**  A $d$-dimensional *Riemannian manifold* $(\mathcal{M}, g)$ is a $d$-dimensional smooth manifold equipped with a Riemannian metric $g$, which assigns to each point $p \in \mathcal{M}$ an inner product

$$g_p : T_p\mathcal{M} \times T_p\mathcal{M} \to \mathbb{R},$$

where $T_p\mathcal{M}$ denotes the tangent space at $p \in \mathcal{M}$. We also write $\langle \cdot, \cdot \rangle_p$ to represent $g_p$ and $\| \cdot \|_p$ for the norm it induces. Let $\phi : \mathcal{M} \to \mathbb{R}^n$ be an embedding from the smooth manifold $\mathcal{M}$ to the Euclidean space $\mathbb{R}^n$. Then $\mathcal{M}$ inherits a Riemannian metric from the ambient Euclidean structure via the pullback metric

$$g_p^E(v, u) := \langle d\phi|_p(v), d\phi|_p(u) \rangle = \langle \phi(v), \phi(u) \rangle,$$

where $\langle \cdot, \cdot \rangle$ denotes the Euclidean inner product on $\mathbb{R}^n$. In this case, we say the metric $g^E$ is induced by the embedding $\phi$, and refer to $\mathbb{R}^n$ as the ambient Euclidean space. To distinguish between Riemannian metrics that may be induced by embeddings into different ambient spaces, we introduce the following definition:

**Definition B.1** ($n$-Euclidean metric). *A Riemannian metric $g$ is called $n$-Euclidean if there exists a smooth embedding $\phi : \mathcal{M} \to \mathbb{R}^n$ such that $g$ is induced by $\phi$.*

Notably, given an arbitrary $d$-dimensional Riemannian manifold $(\mathcal{M}, g)$, the Nash embedding theorem (Nash, 1956; Lee, 2018) states that there always exists $n \in \mathbb{N}$ such that the Riemannian metric $g$ is $n$-Euclidean. However, if we consider a different Riemannian metric $g'$ on the same manifold $\mathcal{M}$, there is no guarantee that $g'$ can also be realized as an $n$-Euclidean metric for the same $n$. This observation motivates us to develop an intrinsic analysis framework that does not depend on any specific embedding.

**Geodesic**    A *vector field* on $\mathcal{M}$ is a smooth *section* $X : \mathcal{M} \to T\mathcal{M}$ of the canonical tangent-bundle projection $\pi : T\mathcal{M} \to \mathcal{M}$; equivalently, it is a smooth map satisfying $\pi \circ X = \mathrm{id}_\mathcal{M}$. Let $\mathfrak{X}(\mathcal{M})$ be the space of all vector fields on a Riemannian manifold $(\mathcal{M}, g)$. The *Levi-Civita connection* is the unique affine connection

$$\nabla : \mathfrak{X}(\mathcal{M}) \times \mathfrak{X}(\mathcal{M}) \to \mathfrak{X}(\mathcal{M}), \quad (X, Y) \mapsto \nabla_X Y,$$

satisfying torsion-free and metric-compatible[2]. Let $I \subset \mathbb{R}$ be an open interval containing $0$. A smooth curve $\gamma : I \to \mathcal{M}$ is called a *geodesic* over $I$ if its velocity vector $\gamma'(t) := d\gamma \mid_t \left( \frac{\partial}{\partial t} \right) \in T_{\gamma(t)}\mathcal{M}$ satisfies the geodesic equation[3]:

$$\nabla_{\gamma'(t)} \gamma'(t) = 0$$

for all $t \in I$. Given a point $p \in \mathcal{M}$ and an initial velocity $v \in T_p\mathcal{M}$, there always exists a unique geodesic $\gamma$ such that $\gamma(0) = p$ and $\gamma'(0) = v$ (Theorem 4.10, Lee (2018)). The *exponential map* at $p$, denoted $\exp_p : T_p\mathcal{M} \to \mathcal{M}$, is defined by $\exp_p(v) := \gamma(1)$. Importantly, the existence of geodesic does not guarantee that $\gamma$ can be defined over an open interval containing $[0, 1]$; that is, the exponential map can be undefined for some $(p, v) \in T\mathcal{M}$. We summarize this observation in the following proposition:

**Proposition B.2** (Proposition 5.7, Lee (2018)). *The exponential map $\exp_p : T_p\mathcal{M} \to \mathcal{M}$ is locally defined on an open neighbor of $0 \in T_p\mathcal{M}$.*

*Remark.* This proposition reveals a fundamental difference between Riemannian and Euclidean zeroth-order optimization: in the Riemannian setting, one cannot simply apply a small perturbation in the direction $v$ at the point $p \in \mathcal{M}$, since the exponential map $\exp_p(\mu v)$ may be undefined. Developing a zeroth-order gradient estimator that operates within this local geometric structure is one of the central goals of our work.

Computing $\exp_p(v)$ involves solving a differentiable equation, which is often costly or intractable; hence, existing Riemannian optimization literature typically uses the first-order approximation called the *retraction* to approximate the exponential map.

**Definition B.3** (Retraction). *A retraction on a manifold $\mathcal{M}$ is a smooth map $\mathrm{Ret} : T\mathcal{M} \to \mathcal{M}$ such that for all $p \in \mathcal{M}$:*

1. *$\mathrm{Ret}_p(0) = p$, where $0 \in T_p\mathcal{M}$ is the zero vector;*

2. *The differential $d\mathrm{Ret}_p|_0 : T_p\mathcal{M} \to T_p\mathcal{M}$ satisfies $d\mathrm{Ret}_p|_0 = \mathrm{id}_{T_p\mathcal{M}}$.*

Here, $\mathrm{Ret}_p : T_p\mathcal{M} \to \mathcal{M}$ denotes the restriction of $R$ to the tangent space at $p$. Intuitively, a retraction approximates $\exp_p(v)$ by preserving the first-order geometry of geodesics while being easier to compute.

The following lemma further characterizes the relation between the exponential map and the retraction. We present it here without providing the proof.

**Lemma B.4** (Theorem 2, Bonnabel (2013)). *Let $(\mathcal{M}, g)$ be a smooth Riemannian manifold.*

---

[2]We call an affine connection torsion-free if $\nabla_X Y - \nabla_Y X = [X, Y]$, where the Lie bracket $[X, Y]$ is defined by $[X, Y](f) = X(Y(f)) - Y(X(f))$ for any $f \in C^\infty(\mathcal{M})$), and metric-compatible if $X\big(g(Y, Z)\big) = g(\nabla_X Y, Z) + g(Y, \nabla_X Z)$ for all $X, Y, Z$.

[3]More explicitly, we choose an extension vector field $\widetilde{X} \in \mathfrak{X}$ satisfying $\widetilde{X}(\gamma(t)) = \gamma'(t)$ for all $t \in I$. Then we define $\nabla_{\gamma'(t)} \gamma'(t) := \nabla_{\widetilde{X}} \widetilde{X} |_{\gamma(t)}$. Here we directly use $\nabla_{\gamma'(t)} \gamma'(t)$ for our convenience, as this definition does not rely on the choice of extension (see Lemma 4.9, Lee (2018)).

   *(i) The exponential map $\exp : T\mathcal{M} \to \mathcal{M}$ is a retraction.*

   *(ii) For every $p \in \mathcal{M}$, the geodesic distance $d(\cdot, \cdot) : \mathcal{M} \times \mathcal{M} \to [0, +\infty)$ between $\exp_p(v)$ and $\mathrm{Ret}_p(v)$ is upper bounded as*

$$d\Big(\exp_p(v), \mathrm{Ret}_p(v)\Big) \le C\|v\|_p^2$$

   *for any $v$ and any retraction $\mathrm{Ret}$.*

**Gradient**   Let the cotangent space $T_p^*\mathcal{M}$ be the dual space of $T_p\mathcal{M}$; that is, the space of all linear mappings $\psi : T_x\mathcal{M} \to \mathbb{R}$. There is a natural isomorphism between $T_p\mathcal{M}$ and $T_p^*\mathcal{M}$ induced by the Riemannian metric $g$:

$$\flat_p : T_p\mathcal{M} \to T_p^*\mathcal{M}, \quad v \mapsto g_p(v, \cdot);$$

$$\sharp_p : T_p^*\mathcal{M} \to T_p\mathcal{M}, \quad \omega \mapsto \omega^\sharp \quad \text{satisfying} \quad g_p(\omega^\sharp, v) = \omega(v),$$

for all $v \in T_p\mathcal{M}$. Let $f : \mathcal{M} \to \mathbb{R}$ be a smooth real-value function. The differential of $f$ at $p \in \mathcal{M}$, given by $df|_p(v) := vf$, naturally defines a covector in the cotangent space; that is, $df|_p \in T_p^*\mathcal{M}$. The gradient of $f$, denoted by $\nabla f \in \mathfrak{X}(\mathcal{M})$, is a vector field given by

$$p \mapsto \nabla f(p) := \big(df|_p\big)^\sharp.$$

In this paper, we investigate the approach of estimating $\nabla f(p)$ given only the access to the function evaluation. There have been a rich body of literature in this direction, which we summarize in Appendix A. In contrast, our approach is purely intrinsic, making our result distinct from existing literature.

## C   MAIN RESULTS

### C.1   ASSUMPTIONS

The following assumption is standard in stochastic optimization literature (Mishchenko et al., 2020; Khaled & Richtárik, 2022). In the context of Riemannian optimization, it is often coupled with Assumption 2.2 to define the $L$-smoothness of the pullback function (Bonnabel, 2013; Li et al., 2023b; He et al., 2024). In contrast, we decouple these two assumptions to make their respective roles and dependencies more transparent.

**Assumption C.1.** *In the optimization problem given by Equation (1), the individual loss function*

$$f(\cdot; \xi) : \mathcal{M} \to \mathbb{R}$$

*satisfies the following two properties:*

   *(a) $L$-Bounded Hessian; for all $p \in \mathcal{M}$, the Hessian matrix at the point $p$ is bounded by $L$.*

   *(b) Lower boundedness; the infimum $f_\xi^* :=$ exists almost surely with $\xi \sim \Xi$.*

The following assumption imposes a regularization condition on the retraction used in Theorem 2.9. While it is always possible to construct a pathological retraction that deviates substantially from the exponential map, such choices may still scale with $\|v\|_p$ but would negatively affect the final convergence rate.

**Assumption C.2.** *Let $f : \mathcal{M} \to \mathbb{R}$ be a smooth function. There exists a constant $C_{\mathrm{Ret}} \ge 0$ such that*

$$|f(\mathrm{Ret}_p(v)) - f(\exp_p(v))| \le C_{\mathrm{Ret}}\|v\|_p^2.$$

*Remark.* This assumption can indeed be replaced with a stronger but more widely used boundedness assumption (*e.g.* the bounded gradient assumption). Bonnabel (2013) has shown that the geodesic distance between the (first-order) retraction $\mathrm{Ret}_p(v)$ and the exponential map $\exp_p(v)$ is of the order $o(\|v\|_p^2)$ (see Theorem 2, Bonnabel (2013)). In Lemma C.11, we show that given appropriate smoothness and boundedness conditions, the gap between $f(\mathrm{Ret}_p(v))$ and $f(\exp_p(v))$ is also of the order $o(\|v\|_p^2)$, which implies Assumption 2.2. Here we present this weaker assumption to avoid introducing the bounded gradient assumption.

**Assumption C.3.** *There exist constants $\rho > 0$ and $M_3, M_4 > 0$ such that*

$$\left\|\nabla^3 f(q)\right\|_{\mathrm{HS}} \leq M_3, \quad \left\|\nabla^4 f(q)\right\|_{\mathrm{HS}} \leq M_4,$$

*for all $q \in \mathcal{B}_p(p, \rho)$, where $\mathcal{B}_p(p, \rho)$ denotes the geodesic ball of radius $\rho$ and $\|\cdot\|_{\mathrm{HS}}$ is the Hilbert-Schmidt norm.*

*Remark.* Unlike the Euclidean setting, optimization on Riemannian manifolds often relies on additional boundedness assumptions. For example, He et al. (2024) and Li et al. (2023b) impose a Lipschitz continuity condition on the Hessian of the pullback objective (Assumption 4.2 in He et al. (2024), Assumption 2.2 in Li et al. (2023b)), which can be viewed as a variant of Assumption 2.3. The assumption of bounded fourth-order derivatives in Assumption 2.3 is less common in the literature. However, we emphasize that it plays a crucial role in our analysis: it enables us to capture the dependence on sectional curvature in the accuracy of zeroth-order gradient estimation (see Theorem 2.7). From our perspective, introducing this assumption leads to a novel and more refined result that has not yet been explored in existing work.

**Assumption C.4.** *There exists a constant $\kappa \geq 0$ such that the sectional curvature of the Riemannian manifold $(\mathcal{M}, g)$ satisfies*

$$|K_p(\sigma)| \leq \kappa, \qquad \textit{for every point } p \in \mathcal{M} \textit{ and every } 2\textit{-plane } \sigma \subset T_p\mathcal{M}.$$

*Equivalently, $-\kappa \leq K_p(\sigma) \leq \kappa$ for all $p$ and $\sigma$.*

*Remark.* Many existing literature (Wang et al., 2021b; 2023) also made assumptions on the sectional curvature (lower) boundedness. Here we present a slightly stronger assumption: we assume the sectional curvature is uniformly bounded (*i.e.* both upper and lower boundedness). We note that this assumption has also been used in existing literature (see Assumption 1, Alimisis et al. (2021)).

## C.2    SUPPORTING LEMMAS

The following lemma generalizes the expected smoothness widely used in non-convex optimization (Mishchenko et al., 2020; Khaled & Richtárik, 2022; Ma & Huang, 2025b;a).

**Lemma C.5.** *Let $f_\xi^* = \inf_{p \in \mathcal{M}} f(p; \xi)$ and $f^* := \inf_p \mathbb{E}_{\xi \sim \Xi} f(\cdot; \xi)$. Suppose that Assumption 2.1 is satisfied and $f^* < +\infty$. Then there exists $A, B \geq 0$ such that for any $p \in \mathcal{M}$,*

$$\mathbb{E}\|\nabla f(p; \xi)\|_p^2 \leq A[f(p) - f^*] + B.$$

*Proof.* Let $\gamma : \mathbb{R} \to \mathbb{R}$ be the geodesic with $\gamma(0) = p$ and $\gamma'(0) = v$ (with the unit length). Then the composite of smooth mappings $f \circ \gamma(\cdot; \xi) : \mathbb{R} \to \mathbb{R}$ is a smooth function. By applying the coercive inequality, i.e. Proposition 2 (Mishchenko et al. (2020)), to $f \circ \gamma(\cdot; \xi)$ (it is equivalent to apply the Taylor formula expanding $f(p; \xi)$ at the minima point $p_\xi^* := \arg\min_p f(p; \xi)$):

$$f(\underbrace{\gamma(0)}_{=p}; \xi) \geq f(p_\xi^*; \xi) + \frac{1}{2L}\|\nabla f(\gamma(0); \xi)\|_p^2,$$

where the inequality we apply the $L$-bounded Hessian assumption (Assumption 2.1). Let $f_\xi^* := f(p_\xi^*; \xi)$ (i.e. $f_\xi^* := \inf_{p \in \mathcal{M}} f(p; \xi)$) be the minima of the individual loss $f(\cdot; \xi)$; then we obtain

$$\begin{aligned} \mathbb{E}_{\xi \sim \Xi}\|\nabla f(p; \xi)\|_p^2 &\leq 2Lf(p) - 2L\mathbb{E}_{\xi \sim \Xi}f_\xi^* \\ &= 2L[f(p) - f^*] + 2L[f^* - \mathbb{E}_{\xi \sim \Xi}f_\xi^*], \end{aligned}$$

where $f^*$ is the minima of the objective loss function. Typically, we have

$$f^* := \inf_p \mathbb{E}_{\xi \sim \Xi} f(\cdot; \xi) \geq \mathbb{E}_{\xi \sim \Xi} f_\xi^* := \mathbb{E}_{\xi \sim \Xi}\left[\inf_p f(\cdot|\xi)\right]$$

by Jensen's inequality using the convexity of the inf operator. Therefore, $B \geq 0$. The proof is completed by defining $A = 2L$ and $B = 2L[f^* - \mathbb{E}_{\xi \sim \Xi}f_\xi^*]$. $\qquad \square$

**Lemma C.6.** *Let $\mathcal{M}$ be a smooth manifold. Then there exists a smooth function $\rho : \mathcal{M} \to [0, +\infty)$ that is proper; i.e., for every compact set $C \subset \mathbb{R}$, $\rho^{-1}(C)$ is compact in $\mathcal{M}$.*

*Proof.* This result directly comes from Proposition 2.28 (Lee, 2003) and it can be directly generalized for arbitrary Hausdorff paracompact topological space, as for a Hausdorff space, the paracompactness is equivalent to the existence of partitions of unity (Dugundji, 1966). Here we present a proof without using the partitions of unity.

By Proposition A.60 (Lee, 2003), the smooth manifold $\mathcal{M}$ admits an exhaustion by compact sets[4]; that is, a sequence of compact sets $\{K_j\}_{j=1}^\infty$ in $\mathcal{M}$, such that

- $K_j \subset K_{j+1}^\circ$ for all $j$;

- $\bigcup_{j=1}^\infty K_j = \mathcal{M}$.

For each $j$, we can always have a smooth function $\psi_j : \mathcal{M} \to [0,1]$ such that $\psi_j \equiv 1$ on $K_j$ and $\mathrm{supp}(\psi_j) \subset K_{j+1}^\circ$. This existence is guaranteed by Proposition 2.25 (Lee, 2003). Define a smooth function $\rho : \mathcal{M} \to [0, +\infty)$ by

$$\rho(p) := \sum_{j=1}^\infty \left(1 - \psi_j(p)\right).$$

For any fixed $p$, there exists a $j$ with $p \in K_j$; as the result, there is at most finite entries in this series non-zero. The finite-sum of smooth functions is also smooth. Moreover, $\rho^{-1}((-\infty, c]) \subset K_{\lfloor c \rfloor + 1}$, which is compact. Since $\rho$ is always non-negative, it implies that $\rho$ is proper. □

*Remark.* If the manifold $\mathcal{M}$ is compact (e.g., a sphere), then every continuous function serves as an exhaustion function. This offers an alternative perspective on the structure-preserving metric: for a compact manifold, we do not need to worry about the exponential map sending points outside the manifold, as all metrics constructed in Theorem C.16 are automatically geodesically complete.

**Lemma C.7** (Isserlis). *Let $(\mathcal{M}, g)$ be a $d$-dimensional smooth Riemannian manifold, $p \in \mathcal{M}$, and $f : \mathcal{M} \to \mathbb{R}$ be a smooth function. Denote by*

$$\partial\mathcal{B} = \left\{v \in T_p\mathcal{M} : \|v\|_g = 1\right\}$$

*the unit sphere in the tangent space. Write $\mathrm{Unif}(\partial\mathcal{B})$ for the corresponding uniform probability measures. If $v = (v_1, v_2, \ldots, v_d) \sim \mathrm{Unif}(\partial\mathcal{B})$ then*

$$\mathbb{E}v_{i_1}v_{i_2}\ldots v_{i_n} = \begin{cases} 0, & 2 \nmid n, \\ \frac{1}{d(d+2)(d+4)\ldots(d+2k-2)} \sum_{pair \in P_{2k}^2} \prod_{(r,s) \in pair} \delta_{i_r, i_s}, & 2 \mid n, \end{cases}$$

*where $P_{2k}^2$ represents the set of all pairings of $\{1, 2, \ldots, 2k\}$ (i.e. all distinct ways of partitioning $\{1, 2, \ldots, n\}$ into pairs $\{r, s\}$), and $\delta_{ij} = \begin{cases} 0 & i \neq j, \\ 1 & i = j, \end{cases}$ is the Kronecker delta.*

*Proof.* This result is known as the generalization of Isserlis's theorem (Isserlis, 1916; 1918). Our presented version is taken from Wikipedia, which refers to Koopmans (1974); Mardia & Jupp (1999). □

**Lemma C.8.** *Let $(\mathcal{M}, g)$ be a $d$-dimensional Riemannian manifold. Assume there exists a constant $\kappa \geq 0$ such that the sectional curvature satisfies*

$$|K_p(\sigma)| \leq \kappa \qquad \text{for every point } p \in \mathcal{M} \text{ and every 2-plane } \sigma \subset T_p\mathcal{M}.$$

*Then, for every $p \in \mathcal{M}$ the Ricci curvature obeys the operator-norm bound*

$$\|\mathrm{Ric}_p\|_{\mathrm{op}} = \sup_{\substack{v \in T_p\mathcal{M} \\ v \neq 0}} \frac{|\mathrm{Ric}_p(v, v)|}{\|v\|_p^2} \leq (d-1)\,\kappa.$$

---

[4]We always require the manifold to be second-countable and Hausdorff; and all topological spaces locally homomorphism to the Euclidean space are locally compact.

*Proof.* Fix a point $p$ and a non-zero vector $v \in T_p\mathcal{M}$. Extend $v$ to an orthonormal basis $\{v/\|v\|_p, e_2, \ldots, e_d\}$ of $T_p\mathcal{M}$. By the classical formula relating Ricci and sectional curvature,

$$\mathrm{Ric}_p(v, v) = \sum_{i=2}^{d} K_p\big(\mathrm{span}\{v, e_i\}\big)\|v\|_p^2.$$

Taking absolute values and using $|K| \le \kappa$ gives

$$|\mathrm{Ric}_p(v, v)| \le (d - 1)\,\kappa\,\|v\|_p^2.$$

Dividing by $\|v\|_p^2$ and taking the supremum over all non-zero $v$ yields $\|\mathrm{Ric}_p\|_{\mathrm{op}} \le (d - 1)\kappa$, as claimed. $\qquad\square$

**Lemma C.9.** *Let $L \in \mathbb{R}^{d \times d}$ be an invertible diffeomorphism defined as*

$$L : \mathbb{S}^{d-1} \to \mathcal{C} := \{v \in \mathbb{R}^d \mid v^\top A v = 1\}, \qquad L(s) = L\,s,$$

*where $L^\top A L = I_d$. Denote by $\sigma_{\mathbb{S}^{d-1}}$ and $\sigma_\mathcal{C}$ the $(d-1)$-dimensional Hausdorff measures on $\mathbb{S}^{d-1}$ and $\mathcal{C}$, respectively. Then $\sigma_{\mathbb{S}^{d-1}} \circ L^{-1}$ is absolutely continuous w.r.t. $\sigma_\mathcal{C}$ and*

$$\frac{d\big(\sigma_{\mathbb{S}^{d-1}} \circ L^{-1}\big)}{d\sigma_\mathcal{C}}(v) = \frac{1}{J\big(L^{-1}v\big)}, \qquad J(s) := |\det L|\,\|(L^\top)^{-1}s\|_2.$$

*Proof.* The result immediately follows Theorem 3.2.3 ([Federer, 1996](#)). Here the linear map $J$ is the $(d-1)$-dimensional Jacobian of $L$ defined as

$$J(s) := J_{d-1}L(s) := \|\bigwedge^{d-1} dL(s)\|_o,$$

where $dL(s) : T_s\mathbb{S}^{d-1} \to T_s\mathcal{C}$ is the differential of $L$, $\bigwedge$ is the wedge product, and $\|\cdot\|_m$ denotes the standard operator norm $\|f\|_o := \sup_{\|x\| \le 1} |f(x)|$. As $L$ is a linear map, the wedge product gives $\bigwedge^{d-1} dL(s) = (\det L)(L^\top)^{-1}$. Taking the norm yields

$$J(s) = |\det L|\|(L^\top)^{-1}s\|_2.$$

Then it completes the proof. $\qquad\square$

**Lemma C.10.** *Let $\gamma$ be a geodesic defined over the open interval $I \ni 0$ satisfying (i) $\gamma(0) = p$ and (ii) $\gamma'(0) = v$. Let $F : I \to \mathbb{R}$ be a scalar function over $I$ defined as*

$$F(t) := \exp_p\big(\gamma(t)\big).$$

*Then the following relations hold:*

*(1) $F'(t) = \nabla f(\gamma(t))[\gamma'(t)]$; $F'(0) = \langle \nabla f(p), v\rangle_p$.*

*(2) $F''(t) = \nabla^2 f(\gamma(t))[\gamma'(t), \gamma'(t)]$; $F''(0) = \nabla^2 f(p)[v, v]$.*

*(3) $F'''(t) = \nabla^3(\gamma(t))[\gamma'(t), \gamma'(t), \gamma'(t)]$; $F'''(0) = \nabla^3 f(p)[v, v, v]$.*

*Proof.* (1) As $F = f \circ \gamma : I \to \mathcal{M} \to \mathbb{R}$, the chain rule gives

$$dF_t = df_{\gamma(t)} \circ d\gamma_t : T_t\mathbb{R} \to T_{\gamma(t)}\mathcal{M} \to T_{f \circ \gamma(t)}\mathbb{R}.$$

We take $\frac{\partial}{\partial t} \in T_t\mathbb{R}$. Then

$$F'(t) := dF_t(\frac{\partial}{\partial t}) = df_{\gamma(t)} \circ \gamma'(t)$$
$$\stackrel{(i)}{=} [\nabla f(\gamma(t))]^\flat \big(\gamma'(t)\big)$$
$$= \langle \nabla f(\gamma(t)), \gamma'(t)\rangle_{\gamma(t)},$$

where (i) applies the isomorphism between $T_p\mathcal{M}$ and $T_p^*\mathcal{M}$ given by $\flat$. When treating $\nabla f(\gamma(t))$ as an element in $T_p^*\mathcal{M}$ through this isomorphism, we also write:

$$\nabla f(\gamma(t))[\gamma'(t)] := [\nabla f(\gamma(t))]^\flat \big(\gamma'(t)\big).$$

Here, we use $\nabla f(p)[\cdot]$ to represent that the gradient $\nabla f(p)$ is understood as a 1-form mapping from $T_p\mathcal{M}$ to $\mathbb{R}$. When $t = 0$, we immediately obtain $F'(0) = \langle \nabla f(p), v\rangle_p$ by using $\gamma(0) = p$ and $\gamma'(0) = v$.

(2) The chain rule gives

$$d^2 F_t = d^2 f_{\gamma(t)}(d\gamma_t, d\gamma_t) + df_{\gamma(t)}(d^2\gamma_t) : T_t\mathbb{R} \times T_t\mathbb{R} \to T_{f\circ\gamma(t)}\mathbb{R}.$$

We take $\frac{\partial}{\partial t} \in T_t\mathbb{R}$. Then

$$F''(t) = d^2 F_t(\frac{\partial}{\partial t}, \frac{\partial}{\partial t}) = d^2 f_{\gamma(t)}(\gamma'(t), \gamma'(t)) + df_{\gamma(t)}(\nabla_{\gamma'(t)}\gamma'(t)).$$

As $df_{\gamma(t)} : T_{\gamma(t)}\mathcal{M} \to T_{f\circ\gamma(t)}\mathbb{R} \cong \mathbb{R}$ is a linear function, it always maps $0$ to $0$. By the property of geodesic, $\nabla_{\gamma'(t)}\gamma'(t) = 0$, leading to

$$F''(t) = d^2 f_{\gamma(t)}(\gamma'(t), \gamma'(t)) = \nabla^2 f(\gamma(t))[\gamma'(t), \gamma'(t)]$$

Here, we directly take $d^2 f_{\gamma(t)} = \nabla^2 f(\gamma'(t))$ as it has been a 2-form in $T^*_{\gamma(t)}\mathcal{M} \otimes T^*_{\gamma(t)}\mathcal{M}$. To align the same notation used in $\nabla$, we still use $[\cdot, \cdot]$. When $t = 0$, we immediately obtain $F''(0) = \nabla^2 f(p)[v, v]$ by using $\gamma(0) = p$ and $\gamma'(0) = v$.

(3) The chain rule gives

$$d^3 F_t = d^3 f_{\gamma(t)}(d\gamma_t, d\gamma_t, d\gamma_t) + 3d^2 f_{\gamma(t)}(d\gamma_t, d^2\gamma_t) + df_{\gamma(t)} \circ d^3\gamma_t.$$

We take $\frac{\partial}{\partial t} \in T_t\mathbb{R}$. As $\gamma : I \to \mathcal{M}$ is a geodesic, the last two terms are zeros. Then

$$F'''(t) = d^3 f_{\gamma(t)}(\gamma'(t), \gamma'(t), \gamma'(t)) := \nabla^3 f(\gamma(t))[\gamma'(t), \gamma'(t), \gamma'(t)].$$

Now the proof is completed. $\qquad\square$

**Lemma C.11.** *Let $f : \mathcal{M} \to \mathbb{R}$ be a smooth function. Suppose that Assumption 2.1 holds. If $\|\nabla f(p)\|_p$ is uniformly bounded by a constant $G > 0$ for all $p \in \mathcal{M}$, then there exists a constant $C_{\text{Ret}} \geq 0$ such that*

$$|f(\text{Ret}_p(v)) - f(\exp_p(v))| \leq C_{\text{Ret}}\|v\|_p^2.$$

*Proof.* It suffices to apply the standard Taylor formula (Spivak, 1994) to both functions

$$f \circ \text{Ret}_p : T_p\mathcal{M} \cong \mathbb{R}^d \to \mathbb{R} \quad \text{and} \quad f \circ \exp_p : \mathbb{R}^d \to \mathbb{R},$$

then evaluate their difference. We set $\gamma(t) := \exp_p(tv)$ as the geodesic and $\gamma_{\text{Ret}}(t) := \text{Ret}_p(tv)$ as the first-order approximation of the geodesic. The Taylor formula gives

$$f \circ \exp_p(v) = f(p) + \langle \nabla f(p), v \rangle_p + \int_0^1 (1-t)\nabla^2 f(\exp_p(tv))[\gamma'(tv), \gamma'(tv)]dt,$$

$$f \circ \text{Ret}_p(v) = f(p) + \langle \nabla f(p), v \rangle_p + \int_0^1 (1-t)\nabla^2 f(\text{Ret}_p(tv))[\gamma'_{\text{Ret}}(t), \gamma'_{\text{Ret}}(t)]dt + \iota,$$

where $\iota$ is the correction term reflecting the curvature from the approximated geodesic $\gamma_{\text{Ret}}$, given by

$$\iota := \int_0^1 (1-t)\langle \nabla f(\gamma_{\text{Ret}}(t)), \nabla_{\gamma'_{\text{Ret}}(t)}\gamma'_{\text{Ret}}(t) \rangle_{\gamma_{\text{Ret}}(t)}dt.$$

When $\text{Ret} \equiv \exp$, the Levi-Civita connection $\nabla : \mathfrak{X}(\mathcal{M}) \times \mathfrak{X}(\mathcal{M}) \to \mathfrak{X}(\mathcal{M})$ automatically gives $\nabla_{\gamma'_{\text{Ret}}(tv)}\gamma'_{\text{Ret}}(tv) = 0$, which recovers the zero approximation error.

When considering a general first-order retraction, we can further upper bound it using the bounded gradient assumption. Since the gradient $\nabla f(\gamma_{\text{Ret}}(t))$ is uniformly bounded, we can also upper bound its directional derivative $\left\|\nabla_{\gamma'_{\text{Ret}}(t)}\gamma'_{\text{Ret}}(t)\right\|_{\gamma_{\text{Ret}}(t)}$; here we set this uniform upper bound as $\ell$.

$$|\iota| = \left| \int_0^1 (1-t)\langle \nabla f(\gamma_{\text{Ret}}(t)), \nabla_{\gamma'_{\text{Ret}}(t)}\gamma'_{\text{Ret}}(t) \rangle_{\gamma_{\text{Ret}}(t)}dt \right|$$

$$\overset{(i)}{\leq} \left| \int_0^1 (1-t)\left\|\nabla f(\gamma_{\text{Ret}}(t))\right\|_{\gamma_{\text{Ret}}(t)} \left\|\nabla_{\gamma'_{\text{Ret}}(t)}\gamma'_{\text{Ret}}(t)\right\|_{\gamma_{\text{Ret}}(t)} dt \right|$$

$$\overset{(ii)}{\leq} \Big| \int_0^1 (1-t) G\ell \|v\|_p^2 \, dt \Big| \leq \frac{G\ell}{2} \|v\|_p^2,$$

where (i) applies the Cauchy–Schwarz inequality, and (ii) applies the uniformly bounded gradient of $f : \mathcal{M} \to \mathbb{R}$ and the uniformly bounded Hessian of $f \circ \mathrm{Ret} : \mathcal{M} \to \mathbb{R}$.

We take the difference of the above two equations. The bounded Hessian assumption implies

$$\big| f \circ \mathrm{Ret}_p(v) - f \circ \exp_p(v) \big| \leq \frac{1}{2} L \|v\|_p^2 + \frac{1}{2} L \|v\|_p^2 + \iota$$

$$\leq (L + \frac{G\ell}{2}) \|v\|_p^2.$$

we obtain the final upper bound by setting $C_{\mathrm{Ret}} = L + \frac{G\ell}{2}$. $\qquad\square$

**Lemma C.12.** *Let $(\mathcal{M}, g)$ be a smooth, $d$-dimensional, geodesically complete Riemannian manifold and let $f : \mathcal{M} \to \mathbb{R}$ be a smooth function. Suppose that Assumption 2.1 and Assumption 2.2 hold. Given a unit-length vector $v \in T_p\mathcal{M}$ and the perturbation stepsize $\mu > 0$, define*

$$\tilde{h} := \frac{f \circ \exp_p(\mu v) - f \circ \exp_p(-\mu v)}{2\mu} v, \qquad \hat{h} := \frac{f \circ \mathrm{Ret}_p(\mu v) - f \circ \mathrm{Ret}_p(-\mu v)}{2\mu} v.$$

*Then*

$$\|\hat{h} - \tilde{h}\|_p \leq C_{\mathrm{Ret}} \mu.$$

*Proof.* We directly take the difference bewtween two vectors:

$$\begin{aligned}
\Big\| \hat{h} - \tilde{h} \Big\|_p &= \Big\| \frac{f \circ \exp_p(\mu v) - f \circ \exp_p(-\mu v)}{2\mu} v - \frac{f \circ \mathrm{Ret}_p(\mu v) - f \circ \mathrm{Ret}_p(-\mu v)}{2\mu} v \Big\|_p \\
&\overset{(i)}{=} \big| f \circ \exp_p(\mu v) - f \circ \exp_p(-\mu v) - f \circ \mathrm{Ret}_p(\mu v) + f \circ \mathrm{Ret}_p(-\mu v) \big| / (2\mu) \\
&\overset{(ii)}{\leq} \big| f \circ \exp_p(\mu v) - f \circ \mathrm{Ret}_p(\mu v) \big| / (2\mu) + \big| f \circ \exp_p(-\mu v) - f \circ \mathrm{Ret}_p(-\mu v) \big| / (2\mu) \\
&\overset{(iii)}{\leq} C_{\mathrm{Ret}} \mu,
\end{aligned}$$

where (i) applies that $v \in T_p\mathcal{M}$ is the unit-length, (ii) applies the triangle inequality, and (iii) applies Assumption 2.2. $\qquad\square$

**Lemma C.13.** *Let $(\mathcal{M}, g)$ be a smooth, $d$-dimensional, geodesically complete Riemannian manifold and let $f : \mathcal{M} \to \mathbb{R}$ be a smooth function. Suppose that Assumptions 2.1 to 2.4 hold. Suppose that there exists a constant $C_{\mathrm{Ret}} \geq 0$ such that*

$$|f(\mathrm{Ret}_p(v)) - f(\exp_p(v))| \leq C_{\mathrm{Ret}} \|v\|_p^2. \tag{6}$$

*Let $\{p_t\}$ be the SGD dynamic solving Equation (1) generated by the update rule Equation (4). Then*

$$\begin{aligned}
\frac{\eta}{6d} \|\nabla f(p_t)\|_{p_t}^2 &\leq \Big[ 1 + 6L(C_{\mathrm{Ret}} + \frac{L}{2})(\frac{2 + \mu^2 \kappa^2}{d})\eta^2 + \frac{L\mu^4 d}{(d+2)^2} \kappa^2 \eta \Big] \Big( \mathbb{E} f(p_t) - f^* \Big) \\
&\quad - \Big( \mathbb{E} f(p_{t+1}) - f^* \Big) + (C_{\mathrm{Ret}} + \frac{L}{2}) \Big( 3B(\frac{2 + \mu^2 \kappa^2}{d}) + 3\mathscr{E} + 3C_{\mathrm{Ret}}^2 \mu^2 \Big) \eta^2 \\
&\quad + \frac{\eta d}{2} \mathscr{F} + \frac{3}{4} d\eta \mu^2 C_{\mathrm{Ret}}^2,
\end{aligned}$$

*where $\mathscr{E}$ and $\mathscr{F}$ are given by Equation (7) and Equation (10), respectively.*

*Proof.* Let $\hat{h}_t = \widehat{\nabla} f(p_t; \xi_t) := \frac{f\big( \mathrm{Ret}_{p_t}(\mu v) \big) - f\big( \mathrm{Ret}_{p_t}(-\mu v) \big)}{2\mu} v \in T_{p_t}\mathcal{M}$ (also defined in Equation (3)), $\tilde{h}_t = \frac{f\big( \exp_{p_t}(\mu v) \big) - f\big( \exp_{p_t}(-\mu v) \big)}{2\mu} v$, and $h_t = \frac{1}{d} \nabla f(p_t; \xi_t) \in T_{p_t}\mathcal{M}$. At the $t$-th update, the SGD update rule (Equation (4)) gives

$$p_{t+1} = \mathrm{Ret}_{p_t}(-\eta \hat{h}_t).$$

Let $\gamma : I \to \mathcal{M}$ be the geodesic over $I \supset [0, 1]$ that satisfies $\gamma(0) = p_t$ with the initial velocity $\gamma'(0) = -\eta \hat{h}_t$. The Taylor formula of the scalar function $f \circ \gamma$ gives

$$f(p_{t+1}) = f\big(\mathrm{Ret}_{p_t}(-\eta \hat{h}_t)\big) - f\big(\exp_{p_t}(-\eta \hat{h}_t)\big) + f\big(\exp_{p_t}(-\eta \hat{h}_t)\big)$$

$$\overset{(i)}{\leq} C_{\mathrm{Ret}}\eta^2 \|\hat{h}_t\|_{p_t}^2 + f(p_t) - \eta\langle \nabla f(p_t), \hat{h}_t - \tilde{h}_t + \tilde{h}_t\rangle_{p_t}$$

$$\qquad + \int_0^1 (1-t)\nabla^2 f(\gamma(t))[\gamma'(t), \gamma'(t)]dt$$

$$\overset{(ii)}{\leq} C_{\mathrm{Ret}}\eta^2 \|\hat{h}_t\|_{p_t}^2 + f(p_t) - \eta\langle \nabla f(p_t), \hat{h}_t - \tilde{h}_t\rangle_{p_t} - \eta\langle \nabla f(p_t), \tilde{h}_t\rangle_{p_t} + \frac{L\eta^2}{2}\|\hat{h}_t\|_{p_t}^2$$

$$\mathbb{E}_{p_t} f(p_{t+1}) \overset{(iii)}{\leq} (C_{\mathrm{Ret}} + \frac{L}{2})\eta^2 \mathbb{E}_{p_t}\|\hat{h}_t\|_{p_t}^2 + f(p_t) - \frac{\eta}{d}\|\nabla f(p_t)\|_{p_t}^2$$

$$\qquad - \eta\langle \nabla f(p_t), \mathbb{E}_{p_t}\tilde{h}_t - h_t\rangle_{p_t} + \eta\|\nabla f(p_t)\|_{p_t}\mathbb{E}_{p_t}\|\hat{h}_t - \tilde{h}_t\|_{p_t}$$

$$\leq (C_{\mathrm{Ret}} + \frac{L}{2})\eta^2\|\hat{h}_t\|_{p_t}^2 + f(p_t) - \frac{\eta}{6d}\|\nabla f(p_t)\|_{p_t}^2 + \frac{\eta d}{2}\|\mathbb{E}_{p_t}\tilde{h}_t - h_t\|_{p_t}^2$$

$$\qquad + \frac{3}{4}d\eta\mu^2 C_{\mathrm{Ret}}^2$$

where (i) applies Equation (6) and the Taylor formula, (ii) applies the bounded Hessian assumption (Assumption 2.1), and (iii) takes the expectation conditional on $p_t$ on both sides; here we use $\mathbb{E}_{p_t}[\cdot]$ to represent $\mathbb{E}[\cdot \mid p_t]$ for convenience. The last step applies $2\langle \alpha u, \frac{1}{\alpha}v\rangle \leq \alpha^2\|u\|^2 + \frac{1}{\alpha^2}\|v\|^2$ for $\alpha > 0$ and Lemma C.12. Then it suffices to upper bound the variance term $\mathbb{E}_{p_t}\|\hat{h}_t\|_{p_t}^2$ and the bias term $\|\mathbb{E}_{p_t}\hat{h}_t - h_t\|_{p_t}^2$.

- **Bounding $\mathbb{E}_{p_t}\|\hat{h}_t\|_{p_t}^2$:** First, we split it following the standard routine,

$$\mathbb{E}_{p_t}\|\hat{h}_t\|_{p_t}^2 = \mathbb{E}_{p_t}\|\hat{h}_t - \tilde{h}_t + \tilde{h}_t - h_t + h_t\|_{p_t}^2$$

$$\qquad \leq 3\mathbb{E}_{p_t}\|\hat{h}_t - \tilde{h}_t\|_{p_t}^2 + 3\mathbb{E}_{p_t}\|\tilde{h}_t - h_t\|_{p_t}^2 + 3\mathbb{E}_{p_t}\|h_t\|_{p_t}^2.$$

  ○ The first term $3\mathbb{E}_{p_t}\|\hat{h}_t - \tilde{h}_t\|_{p_t}^2$ is given by Lemma C.12:

$$3\mathbb{E}_{p_t}\|\hat{h}_t - \tilde{h}_t\|_{p_t}^2 \leq 3C_{\mathrm{Ret}}^2\mu^2$$

  ○ The second term $3\mathbb{E}_{p_t}\|\tilde{h}_t - h_t\|_{p_t}^2$ is given by Theorem C.16:

$$3\mathbb{E}_{p_t}\|\tilde{h}_t - h_t\|_{p_t}^2 \leq 3\frac{1 + \mu^2\kappa^2}{d}\mathbb{E}_{p_t}\big\|\nabla f(p_t; \xi_t)\big\|_{p_t}^2 + 3\mathscr{E}$$

$$\qquad \leq 3\frac{1 + \mu^2\kappa^2}{d}\left[2L\left(f(p_t) - f^*\right) + B\right] + 3\mathscr{E}$$

  where the second inequality applies Lemma C.5 and

$$\mathscr{E} := \mu^2\left[\frac{4}{3}\frac{M_3^2}{d^3} + \frac{M_4^2\mu^4}{288}\right]. \tag{7}$$

  ○ The last term is upper bounded by Lemma C.5:

$$3\mathbb{E}_{p_t}\|h_t\|_{p_t}^2 = \frac{3}{d^2}\mathbb{E}_{p_t}\|\nabla f(p_t; \xi_t)\|_{p_t}^2$$

$$\qquad \leq \frac{6L}{d^2}[f(p_t) - f^*] + \frac{3B}{d^2}$$

Putting all together, we obtain

$$\mathbb{E}_{p_t}\|\hat{h}_t\|_{p_t}^2$$

$$\leq 3\frac{1 + \mu^2\kappa^2}{d}\left[2L\left(f(p_t) - f^*\right) + B\right] + 3\mathscr{E} + 3C_{\mathrm{Ret}}^2\mu^2 + \frac{6L}{d^2}[f(p_t) - f^*] + \frac{3B}{d^2}$$

$$= 6L(\frac{1 + \mu^2\kappa^2}{d} + \frac{1}{d^2})[f(p_t) - f^*] + 3B(\frac{1 + \mu^2\kappa^2}{d} + \frac{1}{d^2}) + 3\mathscr{E} + 3C_{\text{Ret}}^2\mu^2.$$

As $d \geq 1$, we obtain

$$\mathbb{E}_{p_t}\|\hat{h}_t\|_{p_t}^2 \leq 6L(\frac{2 + \mu^2\kappa^2}{d})[f(p_t) - f^*] + 3B(\frac{2 + \mu^2\kappa^2}{d}) + 3\mathscr{E} + 3C_{\text{Ret}}^2\mu^2, \quad (8)$$

where $\mathscr{E}$ is given by [Equation (7)] and $B$ is given by [Lemma C.5].

- **Bounding** $\|\mathbb{E}_{p_t}\tilde{h}_t - h_t\|_{p_t}^2$: Following the same proof as [Theorem C.16], we obtain the expansion of the zeroth-order gradient estimator given by [Equation (14)]. We multiply $v$ on both sides and take the expectation:

$$\mathbb{E}_{p_t}\tilde{h}_t - h_t = \frac{\mu^2}{6d(d+2)}\Big[\nabla(\Delta f)(p_t;\xi_t) + 3\text{Ric}(\cdot,\cdot)\nabla f(p_t;\xi_t)\Big] + \frac{\mu^3}{12}\mathbb{E}\Big[(\mathcal{I}_+ - \mathcal{I}_-)v\Big].$$

Then we take the squared norm to obtain the bias upper bound:

$$\|\mathbb{E}_{p_t}\tilde{h}_t - h_t\|_{p_t}^2 \leq \frac{\mu^4}{9d^2(d+2)^2}\Big[\big\|\nabla^3 f(p_t;\xi_t)\big\|_{\text{HS}}^2 + 9\|\text{Ric}(\cdot,\cdot)\nabla f(p_t;\xi_t)\|_{p_t}^2\Big] + \frac{\mu^6}{144}M_4^2$$

$$\overset{(i)}{\leq} \frac{\mu^4 M_3^2}{9d^2(d+2)^2} + \frac{\mu^6}{144}M_4^2 + \frac{\mu^4}{(d+2)^2}\kappa^2\|\nabla f(p_t;\xi_t)\|_{p_t}^2$$

$$\leq \frac{\mu^4 M_3^2}{9d^2(d+2)^2} + \frac{\mu^6}{144}M_4^2 + \frac{\mu^4}{(d+2)^2}\kappa^2\left[2L\left(f(p_t) - f^*\right) + B\right]$$

$$\leq \frac{2L\mu^4}{(d+2)^2}\kappa^2\left(f(p_t) - f^*\right) + \frac{\mu^4 M_3^2}{9d^2(d+2)^2} + \frac{\mu^6}{144}M_4^2 + \frac{\mu^4}{(d+2)^2}\kappa^2 B,$$
$$(9)$$

where (i) applies [Lemma C.8] to upper bound the Ricci curvature by the sectional curvature. For convenience, we set

$$\mathscr{F} := \frac{\mu^4 M_3^2}{9d^2(d+2)^2} + \frac{\mu^6}{144}M_4^2 + \frac{\mu^4}{(d+2)^2}\kappa^2 B. \quad (10)$$

Combine [Equation (8)] and [Equation (9)], we obtain that

$$\frac{\eta}{6d}\|\nabla f(p_t)\|_{p_t}^2 \leq \left[1 + 6L(C_{\text{Ret}} + \frac{L}{2})(\frac{2 + \mu^2\kappa^2}{d})\eta^2 + \frac{L\mu^4 d}{(d+2)^2}\kappa^2\eta\right]\left(\mathbb{E}f(p_t) - f^*\right)$$

$$- \left(\mathbb{E}f(p_{t+1}) - f^*\right) + (C_{\text{Ret}} + \frac{L}{2})\left(3B(\frac{2 + \mu^2\kappa^2}{d}) + 3\mathscr{E} + 3C_{\text{Ret}}^2\mu^2\right)\eta^2$$

$$+ \frac{\eta d}{2}\mathscr{F} + \frac{3}{4}d\eta\mu^2 C_{\text{Ret}}^2,$$

where $\mathscr{E}$ and $\mathscr{F}$ are given by [Equation (7)] and [Equation (10)], respectively. $\qquad\square$

**Lemma C.14.** *Suppose that* $\mathsf{S} \geq 0$. *Let three real-valued sequences* $\{\theta_t\}_{t=1}^T$, $\{\delta_t\}_{t=1}^{T+1}$, *and* $\{G_t\}_{t=1}^T$ *satisfy*

$$\theta_t \leq (1 + \mathsf{S})\,\delta_t - \delta_{t+1} + G_t,$$

*for all* $1 \leq t \leq T$. *Then the iterate is bounded by*

$$\min_{1 \leq t \leq T}\theta_t \leq \frac{\mathsf{S}(1+\mathsf{S})^T}{(1+\mathsf{S})^T - 1}\,\delta_1 + \max_{1 \leq t \leq T}G_t \leq \frac{e^{\mathsf{S}}}{T}\delta_1 + \max_{1 \leq t \leq T}G_t.$$

*Proof.* We telescope the iterative relation by using

$$\theta_T \leq (1 + \mathsf{S})\delta_T - \delta_{T+1} + G_T$$

$$(1 + \mathsf{S}) \times \theta_{T-1} \leq (1 + \mathsf{S})^2\delta_{T-1} - (1 + \mathsf{S})\delta_T + (1 + \mathsf{S})G_{T-1}$$

$$\vdots$$

$$(1 + \mathsf{S})^{T-1} \times \theta_1 \leq (1 + \mathsf{S})^T \delta_1 - (1 + \mathsf{S})^{T-1} \delta_2 + (1 + \mathsf{S})^{T-1} G_1.$$

We sum them together and obtain

$$\left[ \sum_{i=0}^{T-1} (1 + \mathsf{S})^i \right] \min_{1 \leq t \leq T} \theta_t \leq (1 + \mathsf{S})^T \delta_1 + \left[ \sum_{i=0}^{T-1} (1 + \mathsf{S})^i \right] \max_{1 \leq t \leq T} G_t.$$

Then we re-arrange the above inequality and obtain

$$
\begin{aligned}
\min_{1 \leq t \leq T} \theta_t &\leq \frac{(1 + \mathsf{S})^T}{\sum_{i=0}^{T-1} (1 + \mathsf{S})^i} \delta_1 + \max_{1 \leq t \leq T} G_t \\
&= \frac{\mathsf{S}(1 + \mathsf{S})^T}{(1 + \mathsf{S})^T - 1} \delta_1 + \max_{1 \leq t \leq T} G_t \\
&\overset{(i)}{\leq} \frac{e^{\mathsf{S}T}}{T} \delta_1 + \max_{1 \leq t \leq T} G_t,
\end{aligned}
$$

where (i) applies two inequalities $(1 + x)^T \leq e^{Tx}$ and $(1 + x)^T - 1 \geq Tx$. $\qquad\square$

## C.3 Proof of Theorem 2.6

**Theorem C.15.** *Let $\mathcal{M}$ be a smooth manifold (possibly non-compact), and let $g$ be any Riemannian metric on $\mathcal{M}$. Then there exists a Riemannian metric $g'$ on $\mathcal{M}$ which is structure-preserving with respect to $g$.*

*Proof.* In this proof, we distinguish the norms induced by different Riemannian metrics by explicitly writing $\| \cdot \|_{p,g}$ or $\| \cdot \|_{p,g'}$. Elsewhere in the paper, we simply use $\| \cdot \|_p$, as no alternative metric is under consideration.

We mainly follow the construction given by Nomizu & Ozeki (1961) to obtain a conformally equivalent Riemannian metric which is geodesically complete. By Lemma C.6, there exists a smooth proper function $\rho : \mathcal{M} \to [0, +\infty)$. Define the conformal coefficient $h : \mathcal{M} \to (0, +\infty)$ as

$$h(p) := \left( \| \nabla \rho(p) \|_p^2 + 1 \right)^\vartheta,$$

where $\nabla \rho(p) \in T_p \mathcal{M}$ is the gradient of $\rho$ at $p \in \mathcal{M}$ and $\vartheta \geq 1$. Then we define the conformal metric $g'$ as

$$g_p'(v, w) := h(p) g_p(v, w).$$

Now we turn to prove that $(\mathcal{M}, g')$ is a complete metric space; that is, every Cauchy sequence is convergent. Let $\gamma : [a, b] \to \mathcal{M}$ be a piecewise smooth curve segment. Then the length of $\gamma$ with respect to the metric $g'$ is given by

$$
\begin{aligned}
L_{g'}(\gamma) &= \int_a^b \sqrt{g_{\gamma(t)}'(\gamma'(t), \gamma'(t))} \, dt \\
&= \int_a^b \sqrt{h(\gamma(t)) g_{\gamma(t)}(\gamma'(t), \gamma'(t))} \, dt \\
&= \int_a^b \sqrt{h(\gamma(t))} \| \gamma'(t) \|_{\gamma(t), g} \, dt \\
&\overset{(i)}{=} \int_a^b \sqrt{(\| \nabla \rho(\gamma(t)) \|_{\gamma(t), g}^2 + 1)^\vartheta} \| \gamma'(t) \|_{\gamma(t), g} \, dt \\
&\geq \int_a^b \| \nabla \rho(\gamma(t)) \|_{\gamma(t), g} \| \gamma'(t) \|_{\gamma(t), g} \, dt \\
&\overset{(ii)}{\geq} \int_a^b \left| g_{\gamma(t)} \langle \nabla \rho(\gamma(t)), \gamma'(t) \rangle \right| dt \\
&= \int_a^b \left| d\rho_{\gamma(t)}(\gamma'(t)) \right| dt
\end{aligned}
$$

$$\geq \left| \int_a^b d\rho_{\gamma(t),g}(\gamma'(t))\, dt \right|$$

$$= |\rho(\gamma(b)) - \rho(\gamma(a))|,$$

where (i) applies the definition of $h$, and (ii) applies the Cauchy-Schwarz inequality. As a result, for arbitrary $p, q \in \mathcal{M}$, we have

$$|\rho(p) - \rho(q)| \leq d_{g'}(p, q). \tag{11}$$

Let $\{p_k\} \subset \mathcal{M}$ be a Cauchy sequence with respect to $g'$. Then Equation (11) implies that $\{\rho(p_k)\} \subset \mathbb{R}$ must be a Cauchy sequence. We can take a finite supremum

$$c := \sup_k \rho(p_k) < +\infty.$$

Then $\{p_k\} \subset \rho^{-1}([0, c])$; that is, every Cauchy sequence belongs to a compact set by our construction (Lemma C.6), which implies the completeness of $(\mathcal{M}, g')$.

The Hopf-Rinow theorem (Hopf & Rinow, 1931; do Carmo, 1992) states that for a connected Riemannian manifold, geodesic completeness is equivalent to the metric completeness. As we have shown that the *conformally equivalent* metric $g'_p := h(p)g_p$ induces a complete metric space, it automatically makes $(\mathcal{M}, g')$ a geodesically complete manifold. If $\mathcal{M}$ is not connected, this argument applies to each connected component, and a geodesic is contained within a single component. Thus, $(\mathcal{M}, g')$ is geodesically complete.

Lastly, we show that if the $\epsilon$-stationary point under $g$ also gives an $\epsilon$-stationary point under $g'$. Recall that we always have

$$g_p(\nabla_g f(p), v) = df_p(v) = g'_p(\nabla_{g'} f(p), v)$$

for all $v \in T_p\mathcal{M}$. Then

$$h(p)g_p(\nabla_{g'} f(p), v) = g_p(\nabla_g f(p), v).$$

As it holds for all $v$ and $g_p$ is a bilinear form over the linear space $T_p\mathcal{M}$, we obtain

$$h(p)\nabla_{g'} f(p) = \nabla_g f(p).$$

Suppose that $\|\nabla_g f(p)\|_{p,g} \leq \epsilon$, then

$$\begin{aligned}
\|\nabla_{g'} f(p)\|_{p,g'} &= \sqrt{g'_p(\nabla_{g'} f(p), \nabla_{g'} f(p))} \\
&= \sqrt{1/h(p)}\sqrt{g_p(\nabla_g f(p), \nabla_g f(p))} \\
&= \sqrt{1/h(p)}\|\nabla_g f(p)\|_{p,g} \\
&= \sqrt{\frac{1}{\left(\|\nabla\rho(p)\|_p^2 + 1\right)^\vartheta}}\|\nabla_g f(p)\|_{p,g} \\
&\leq \|\nabla_g f(p)\|_{p,g} \leq \epsilon.
\end{aligned}$$

Therefore, we complete the proof. $\qquad \square$

## C.4 Proof of Theorem 2.7

In this subsection, we provide the proof for Theorem 2.7.

**Theorem C.16.** *Let $(\mathcal{M}, g)$ be a complete $d$-dimensional Riemannian manifold and $p \in \mathcal{M}$. Let $f : \mathcal{M} \to \mathbb{R}$ be a smooth function and suppose that Assumptions 2.3 and 2.4 hold. Fix a perturbation stepsize $\mu > 0$ satisfying*

$$\mu^2 \leq \min\{\frac{1}{d-1}, \frac{1}{2} + \frac{6}{d} + \frac{8}{d^2}\},$$

*and for any unit vector $v \in T_p\mathcal{M}$ define the symmetric zeroth-order estimator*

$$\widehat{\nabla} f(p; v) := \frac{f\left(\exp_p(\mu v)\right) - f\left(\exp_p(-\mu v)\right)}{2\mu} v.$$

*Then, for $v \sim \mathrm{Unif}(\mathbb{S}^{d-1})$ uniformly sampled from the $g_p$-unit sphere in $T_p\mathcal{M}$,*

$$\mathbb{E}_{v \sim \mathrm{Unif}(\mathbb{S}^{d-1})}\left[\left\|\widehat{\nabla} f(p; v) - \frac{1}{d}\nabla f(p)\right\|_p^2\right] \leq \frac{1 + \mu^2\kappa^2}{d}\|\nabla f(p)\|_p^2 + \mu^2\left[\frac{4}{3}\frac{M_3^2}{d^3} + \frac{M_4^2\mu^4}{288}\right].$$

*Proof.* Let $\gamma(t) := \exp_p(tv)$ be the geodesic; it satisfies (i) $\gamma(0) = p$ and (ii) $\gamma'(0) = v$. For the scalar function $F(t) := f(\gamma(t))$, we apply the ordinary Taylor theorem (with the integral remainder) at $t = 0$ up to order $4$ (Spivak, 1994; Bonnabel, 2013):

$$F(\mu) = F(0) + \mu F'(0) + \mu^2 \frac{F''(0)}{2} + \mu^3 \frac{F'''(0)}{6} + \frac{1}{6} \int_0^\mu (\mu - t)^3 F''''(t) \, dt.$$

By applying Lemma C.10, we obtain

$$f(\gamma(\mu)) = f(p) + \mu \langle \nabla f(p), v \rangle_p + \frac{\mu^2}{2} \nabla^2 f(p)(v, v) + \frac{\mu^3}{6} \nabla^3 f(p)(v, v, v) \tag{12}$$
$$+ \frac{\mu^4}{6} \underbrace{\int_0^1 (1-t)^3 \nabla^4 f(\gamma(\mu t))(\gamma'(\mu t), \gamma'(\mu t), \gamma'(\mu t), \gamma'(\mu t)) dt}_{\mathcal{I}_+},$$

$$f(\gamma(-\mu)) = f(p) - \mu \langle \nabla f(p), v \rangle_p + \frac{\mu^2}{2} \nabla^2 f(p)(v, v) - \frac{\mu^3}{6} \nabla^3 f(p)(v, v, v) \tag{13}$$
$$+ \frac{\mu^4}{6} \underbrace{\int_0^1 (1-t)^3 \nabla^4 f(\gamma(-\mu t))(\gamma'(-\mu t), \gamma'(-\mu t), \gamma'(-\mu t), \gamma'(-\mu t)) dt}_{\mathcal{I}_-},$$

where the $k$-th covariant derivative at $p \in \mathcal{M}$ is a symmetric $k$-linear form in $\underbrace{T_p^* \mathcal{M} \otimes \cdots \otimes T_p^* \mathcal{M}}_{k \text{ copies}}$

$$\nabla^k f(p) : \underbrace{T_p \mathcal{M} \times \cdots \times T_p \mathcal{M}}_{k \text{ copies}} \to \mathbb{R},$$

and we represent the remainder term given by the Taylor theorem as

$$\mathcal{I}_\pm := \int_0^1 (1-t)^3 \nabla^4 f(\gamma(\pm\mu t))(\gamma'(\mu t), \gamma'(\mu t), \gamma'(\mu t), \gamma'(\mu t)) dt.$$

Subtracting Equation (13) from Equation (12) and dividing by $2\mu$ we obtain

$$\frac{f(\exp_p(\mu v)) - f(\exp_p(-\mu v))}{2\mu} = \langle \nabla f(p), v \rangle_p + \frac{\mu^2}{6} \nabla^3 f(p)(v, v, v) + \frac{\mu^3}{12}(\mathcal{I}_+ - \mathcal{I}_-). \tag{14}$$

Multiplying $v$ on both sides, we obtain

$$\widehat{\nabla} f(p; v) = \frac{1}{d} \nabla f(p) + \underbrace{\left( \langle \nabla f(p), v \rangle_p v - \frac{1}{d} \nabla f(p) \right)}_{=:Z_0(v)} + \mu^2 \underbrace{\frac{1}{6} \nabla^3 f(p)(v, v, v) v}_{=:Z_2(v)} + \underbrace{\frac{\mu^3}{12}(\mathcal{I}_+ - \mathcal{I}_-) v}_{=:R(v)}.$$

By defining these shorthand notations, we have the following compact form:

$$\widehat{\nabla} f(p; v) - \frac{1}{d} \nabla f(p) = Z_0(v) + \mu^2 Z_2(v) + R(v).$$

We take squared-norm on both sides and treating $v$ as the uniform distribution over the $g$-unit sphere $\mathbb{S}^{d-1}$ in $T_p \mathcal{M}$. Then we obtain

$$\mathbb{E}_v \| \widehat{\nabla} f(p; v) - \frac{1}{d} \nabla f(p) \|_p^2$$
$$= \mathbb{E}_v \| Z_0(v) \|_p^2 + \mathbb{E}_v \| \mu^2 Z_2(v) + R(v) \|_p^2 + 2 \mathbb{E}_v \langle Z_0(v), \mu^2 Z_2(v) + R(v) \rangle_p$$
$$= \mathbb{E}_v \| Z_0(v) \|_p^2 + \mathbb{E}_v \| \mu^2 Z_2(v) + R(v) \|_p^2 + 2\mu^2 \mathbb{E}_v \langle Z_0(v), Z_2(v) \rangle_p$$
$$\leq \mathbb{E}_v \| Z_0(v) \|_p^2 + 2\mu^4 \mathbb{E}_v \| Z_2(v) \|_p^2 + 2 \mathbb{E}_v \| R(v) \|_p^2 + 2\mu^2 \mathbb{E}_v \langle Z_0(v), Z_2(v) \rangle_p$$
$$\leq (1 + \mu^2) \mathbb{E}_v \| Z_0(v) \|_p^2 + (2\mu^4 + \mu^2) \mathbb{E}_v \| Z_2(v) \|_p^2 + 2 \mathbb{E}_v \| R(v) \|_p^2,$$

The cross term $\langle Z_0(v), R(v) \rangle_p$ is canceled out by Lemma C.7. More explicitly, we have

$$\mathbb{E}\langle Z_0(v), R(v) \rangle_p = \mathbb{E}\frac{\mu^3(\mathcal{I}_+ - \mathcal{I}_-)}{12}\left\langle \langle \nabla f(p), v \rangle_p v - \frac{1}{d}\nabla f(p), v \right\rangle_p$$

$$\overset{(i)}{=} \frac{\mu^3(\mathcal{I}_+ - \mathcal{I}_-)}{12}\left( 0 - 0 \right) = 0,$$

where (i) applies Lemma C.7. Now it suffices to bound each squared term.

1. **Bounding $\mathbb{E}_v\|R(v)\|_p^2$:** By Assumption 2.3 and $\|v\|_p = 1$, we have

$$\left| \mathcal{I}_\pm(\mu, v) \right| \leq \int_0^1 (1-t)^3 M_4 dt = \frac{M_4}{4}.$$

We have the similar upper bound for $|\mathcal{I}_-|$. Then $|\mathcal{I}_+ - \mathcal{I}_-| \leq |\mathcal{I}_+| + |\mathcal{I}_-| \leq \frac{M_4}{4} + \frac{M_4}{4} = \frac{M_4}{2}$. As the result,

$$\left\| R(v) \right\|_p \leq \frac{\mu^3}{12} \cdot \frac{M_4}{2} = \frac{M_4 \mu^3}{24}.$$

Therefore, we obtain

$$\mathbb{E}_v\left[ \|R(v)\|_p^2 \right] \leq \frac{M_4^2 \mu^6}{576}. \tag{15}$$

2. **Bounding $\mathbb{E}_v\|Z_0(v)\|^2$:** Recall that $Z_0(v) = \langle \nabla f(p), v \rangle_p v - \frac{1}{d}\nabla f(p)$. Then

$$\|Z_0(v)\|_p^2 = g\Big( \langle \nabla f(p), v \rangle_p v - \frac{1}{d}\nabla f(p), \langle \nabla f(p), v \rangle_p v - \frac{1}{d}\nabla f(p) \Big)$$

$$= \langle \nabla f(p), v \rangle_p^2 g(v, v) + \frac{1}{d^2}g(\nabla f(p), \nabla f(p)) - \frac{2}{d}\langle \nabla f(p), v \rangle_p g(\nabla f(p), v)$$

$$= \langle \nabla f(p), v \rangle_p^2 \|v\|_p^2 + \frac{1}{d^2}\|\nabla f(p)\|_p^2 - \frac{2}{d}\langle \nabla f(p), v \rangle_p g(\nabla f(p), v)$$

$$\overset{(i)}{=} (1 - \frac{2}{d})\langle \nabla f(p), v \rangle_p^2 + \frac{1}{d^2}\|\nabla f(p)\|_p^2.$$

where (i) applies $\|v\|_p^2 = 1$ and $g(\nabla f(p), v) = \langle \nabla f(p), v \rangle_p$. By the symmetry of the $\|\cdot\|_p$-norm ball, we have

$$\mathbb{E}_v[v \otimes v] = \frac{1}{d}g_p,$$

where $v \otimes v : T_p\mathcal{M} \times T_p\mathcal{M} \to \mathbb{R}$ is the tensor product of the vector $v$ with itself and $v \otimes v(\nabla f(p), \nabla f(p)) = g_p(v, \nabla f(p))^2$. As the result,

$$\mathbb{E}_v\langle \nabla f(p), v \rangle_p^2 = \frac{1}{d}g_p(\nabla f(p), \nabla f(p)) = \frac{1}{d}\|\nabla f(p)\|_p^2.$$

Therefore, we have

$$\mathbb{E}_v\|Z_0(v)\|^2 = (\frac{1}{d} - \frac{1}{d^2})\|\nabla f(p)\|_p^2 \tag{16}$$

3. **Bounding $\mathbb{E}_v\|Z_2(v)\|^2$:** We choose an orthonormal frame $\{e_1, \ldots, e_d\}$ for $T_p\mathcal{M}$ so that every vector $v \in T_p\mathcal{M}$ with $\|v\|_p = 1$ is represented as

$$v = \sum_{i=1}^d v^i e_i$$

and we write its coordinate as $v = (v^1, v^2, \ldots, v^d) \in \mathbb{R}^d$. As $\nabla^3 f(p) \in T_p^*\mathcal{M} \otimes T_p^*\mathcal{M} \otimes T_p^*\mathcal{M}$, we write the tensor representation as

$$T_{ijk} := (\nabla^3 f)_{ijk}(p).$$

Therefore, we obtain

$$Z_2(v) = \frac{1}{6}\nabla^3 f(p)(v, v, v)v = \frac{1}{6}T_{ijk}v^i v^j v^k v^\ell e_\ell,$$

where we use Einstein notation to represent the sum. By the orthonormal frame, we obtain

$$\|Z_2(v)\|_p^2 = \frac{1}{36} T_{ijk} T_{i'j'k'} v^i v^j v^k v^{i'} v^{j'} v^{k'}.$$

Then it suffices to calculate $\mathbb{E}_v[v^i v^j v^k v^{i'} v^{j'} v^{k'}]$. By Lemma C.7, we obtain

$$\mathbb{E}_v[v^i v^j v^k v^{i'} v^{j'} v^{k'}] = \begin{cases} \frac{6}{d(d+2)(d+4)} & \text{if} \quad (i,j,k) = (i',j',k') \\ \frac{9}{d(d+2)(d+4)} & \text{if} \quad i=j, i'=j', k=k' \\ 0 & \text{otherwise} \end{cases}.$$

As the result, we obtain

$$\mathbb{E}_v\big[\|Z_2(v)\|^2\big] = \frac{1}{36d(d+2)(d+4)} \left[6 T_{ijk} T_{ijk} + 9 T_{iik} T_{jjk}\right].$$

Recall that $T_{ijk} T_{ijk} = \|\nabla^3 f(p)\|_{\mathrm{HS}}^2$. We also have

$$\begin{aligned} T_{iik} T_{jjk} &= \left\|\nabla(\Delta f) + \mathrm{Ric}(\cdot,\cdot)\nabla f(p)\right\|_p^2 \\ &\leq 2\|\nabla(\Delta f)\|_p^2 + 2\|\mathrm{Ric}(\cdot,\cdot)\nabla f(p)\|_p^2 \\ &\leq 2\|\nabla^3 f(p)\|_{\mathrm{HS}}^2 + 2\|\mathrm{Ric}(\cdot,\cdot)\nabla f(p)\|_p^2. \end{aligned}$$

As the result, we obtain

$$\mathbb{E}_v\|Z_2(v)\|_p^2 \leq \frac{1}{6d(d+2)(d+4)} \left[4\|\nabla^3 f(p)\|_{\mathrm{HS}}^2 + 3\|\mathrm{Ric}(\cdot,\cdot)\nabla f(p)\|_p^2\right]. \qquad (17)$$

Combining Equations (15) to (17), we obtain

$$\begin{aligned} &\mathbb{E}_v\left[\left\|\widehat{\nabla} f(p;v) - \frac{1}{d}\nabla f(p)\right\|_p^2\right] \\ &\leq \left(\frac{1}{d} - \frac{1}{d^2}\right)\left(1+\mu^2\right)\|\nabla f(p)\|_p^2 + \frac{2\mu^4 + \mu^2}{6d(d+2)(d+4)}\left(4\|\nabla^3 f(p)\|_{\mathrm{HS}}^2 + 3\|\mathrm{Ric}(\cdot,\cdot)\nabla f(p)\|_p^2\right) \\ &\quad + \frac{M_4^2\mu^6}{288} \\ &\overset{(i)}{\leq} \left(\frac{1}{d} - \frac{1}{d^2}\right)\left(1+\mu^2\right)\|\nabla f(p)\|_p^2 + \frac{2\mu^4 + \mu^2}{6d(d+2)(d+4)}\left(4M_3^2 + 3\kappa^2 d^2\|\nabla f(p)\|_p^2\right) + \frac{M_4^2\mu^6}{288} \\ &\overset{(ii)}{\leq} \left[\left(\frac{1}{d} - \frac{1}{d^2}\right)\left(1+\mu^2\right) + 3\kappa^2 d^2 \frac{2\mu^4 + \mu^2}{6d(d+2)(d+4)}\right]\|\nabla f(p)\|_p^2 + \frac{2\mu^4 + \mu^2}{6d(d+2)(d+4)} 4M_3^2 + \frac{M_4^2\mu^6}{288} \end{aligned}$$

where (i) applies Lemma C.8 and assumptions 2.3 and 2.4. Furthermore, we set

$$3\kappa^2 d^2 \frac{2\mu^4 + \mu^2}{6d(d+2)(d+4)} \leq \frac{\kappa^2\mu^2}{d}.$$

It solves

$$\mu^2 \leq \frac{1}{2} + \frac{6}{d} + \frac{8}{d^2}. \qquad (18)$$

We also let

$$\mu^2 \leq \frac{1}{d-1} \qquad (19)$$

We obtain $\left(\frac{1}{d} - \frac{1}{d^2}\right)\left(1+\mu^2\right) \leq \frac{1}{d}$. It concludes that

$$\mathbb{E}_v\left[\left\|\widehat{\nabla} f(p;v) - \frac{1}{d}\nabla f(p)\right\|_p^2\right] \leq \frac{1+\mu^2\kappa^2}{d}\|\nabla f(p)\|_p^2 + \mu^2\left[\frac{4}{3}\frac{M_3^2}{d^3} + \frac{M_4^2\mu^4}{288}\right].$$

Then the proof is completed. Combining Equations (18) and (19) leads to the range of $\mu$. $\qquad \square$

### C.5 PROOF OF PROPOSITION 2.8

**Proposition C.17.** *Let the vector $v$ be generated by Algorithm 1. Then it follows the uniform distribution over the compact set $\mathcal{C} := \{v \in \mathbb{R}^d : v^\top A v = 1\}$.*

*Proof.* Fix a positive definite matrix $A \in \mathbb{R}^{d \times d}$ and consider its eigenvlue decomposition

$$A = Q \Lambda Q^\top, \quad \Lambda = \operatorname{diag}(\lambda_1, \ldots, \lambda_d), \quad 0 < \lambda_1 \leq \cdots \leq \lambda_d = \lambda_{\max}.$$

Recall that $L := Q \Lambda^{-1/2}$. Then

$$\det L = \det Q \det \Lambda^{-1/2} = \left( \prod_{i=1}^d \lambda_i \right)^{-1/2} > 0.$$

We observe that for every $s \in \mathbb{S}^{d-1}$,

$$(Ls)^\top A L s = s^\top L^\top A L s = 1.$$

It indicates that $Ls \in \mathcal{C} := \{v : v^\top A v = 1\}$. As the result, $L$ defines a smooth bijection linear map from the sphere $\mathbb{S}^{d-1}$ to the compact set $\mathcal{C}$:

$$L : \mathbb{S}^{d-1} \to \mathcal{C}, \quad s \mapsto v = Ls.$$

Under this notation, $\mu_{\mathrm{prop}}$, the distribution of the sampled vector $v$ (without rejection) in Algorithm 1 is given by the push-forward distribution of the uniform distribution via the linear map $L$. That is, any measurable $E \subseteq \mathcal{C}$,

$$\mu_{\mathrm{prop}}(E) := \mu_{\mathbb{S}^{d-1}}(L^{-1}(E)) = \mu_{\mathbb{S}^{d-1}} \circ L^{-1}(E), \tag{20}$$

where $\mu_{\mathbb{S}^{d-1}}$ is the uniform distribution over the sphere $\mathbb{S}^{d-1}$.

Denote by $\sigma_{\mathbb{S}^{d-1}}$ and $\sigma_{\mathcal{C}}$ the Hausdorff measures on $\mathbb{S}^{d-1}$ and $\mathcal{C}$, respectively. Then we re-write the above distribution $\mu_{\mathrm{prop}}$ and $\mu_{\mathbb{S}^{d-1}}$ in the density form; that is

$$\mu_{\mathrm{prop}} = \rho_{\mathrm{prop}} d\sigma_{\mathcal{C}},$$
$$\mu_{\mathbb{S}^{d-1}} = \rho_{\mathbb{S}^{d-1}} d\sigma_{\mathbb{S}^{d-1}}.$$

For arbitrary integral function $g : \mathcal{C} \to \mathbb{R}$, we have

$$\begin{aligned}
\int_{\mathcal{C}} g(v) d\mu_{\mathrm{prop}}(v) &= \int_{\mathbb{S}^{d-1}} g(Ls) d\mu_{\mathrm{prop}}(Ls) \\
&\overset{(i)}{=} \int_{\mathbb{S}^{d-1}} g(Ls) d\mu_{\mathbb{S}^{d-1}} \circ L^{-1}(Ls) \\
&= \int_{\mathbb{S}^{d-1}} g(Ls) d\mu_{\mathbb{S}^{d-1}}(s).
\end{aligned}$$

where (i) applies the definition of the pull-back measure $\mu_{\mathrm{prop}}$ (Equation (20)). Then we obtain

$$\begin{aligned}
\int_{\mathcal{C}} g(v) \rho_{\mathrm{prop}}(v) d\sigma_{\mathcal{C}}(v) &= \int_{\mathbb{S}^{d-1}} g(Ls) \rho_{\mathbb{S}^{d-1}}(s) d\sigma_{\mathbb{S}^{d-1}}(s) \\
&\overset{(i)}{=} \int_{\mathbb{S}^{d-1}} g(Ls) \frac{\rho_{\mathbb{S}^{d-1}}(s)}{J(s)} d\sigma_{\mathcal{C}}(Ls).
\end{aligned}$$

where (i) applies Lemma C.9 with $J(s) = |\det L| \, \|(L^\top)^{-1} s\|_2$. As it holds for all measurable function $g$, it solves the density of $\mu_{\mathrm{prop}}$ as

$$\begin{aligned}
\rho_{\mathrm{prop}}(v) &= \frac{\rho_{\mathbb{S}^{d-1}} \circ L^{-1}(v)}{J \circ L^{-1}(v)} \\
&\propto \frac{1}{\|Av\|_2}.
\end{aligned}$$

Then we consider the rejection step and the final density. Let $\rho_{\text{out}}$ be the density of the output vector of [Algorithm 1](#). Recall that [Algorithm 1](#) accepts the candidate $v = L\,s$ with probability

$$a(v) := \mathbb{P}(\text{accept } v|v) = \mathbb{P}(u < \sqrt{\frac{v^\top A^2 v}{\lambda_{\max}}}|v) = \sqrt{\frac{v^\top A^2 v}{\lambda_{\max}}}.$$

The density of the output vector is given as

$$\rho_{\text{out}}(v) \propto \rho_{\text{prop}}(v)a(v) = \frac{1}{\sqrt{\lambda_{\max}}}.$$

As it is a constant over the compact set $\mathcal{C}$, it is the uniform distribution over $\mathcal{C}$. We also note that the acceptance probability is strictly positive; hence, the loop halts almost surely. This completes the proof of [Proposition 2.8](#). $\qquad\square$

## C.6 Proof of Theorem 2.9

In this section, we present the proof of [Theorem 2.9](#). We write $a \lesssim b$ if there exists a constant $\mathsf{C} > 0$ such that $a \le \mathsf{C}\,b$. The hidden constant $\mathsf{C}$ may depend only on fixed problem parameters.

**Theorem C.18.** *Let $(\mathcal{M}, g)$ be a complete $d$-dimensional Riemannian manifold. Let $f : \mathcal{M} \to \mathbb{R}$ be a smooth function and suppose that [Assumptions 2.1 to 2.4](#) hold. Define the symmetric zeroth-order estimator as in [Equation (5)](#). Let $\{p_t\}_{t=1}^T$ be the SGD dynamic finding the stationary point of [Equation (1)](#) generated by the update rule [Equation (4)](#) with requiring $\eta \lesssim \sqrt{\frac{d}{T}}$ and $\mu^2 \lesssim \sqrt{\frac{d}{T}}$ (explicitly specified in [Equation (21)](#)), then there exists constants $\mathsf{C}_1, \mathsf{C}_2, \mathsf{C}_3 > 0$ such that*

$$\min_{1 \le t \le T} \|\nabla f(p_t)\|_{p_t}^2 \le \mathsf{C}_1 \frac{d}{\eta T} + \mathsf{C}_2\,\eta + \mathsf{C}_3\,d^2\mu^2.$$

*In particular, choosing $\mu \lesssim \frac{1}{d^2}\sqrt{\frac{d}{T}}$ yields*

$$\min_{1 \le t \le T} \|\nabla f(p_t)\|_{p_t}^2 \lesssim \sqrt{\frac{d}{T}}.$$

*Proof.* By [Lemma C.13](#), we obtain that

$$\frac{\eta}{6d}\|\nabla f(p_t)\|_{p_t}^2 \le \left[1 + 6L(C_{\text{Ret}} + \frac{L}{2})(\frac{2 + \mu^2\kappa^2}{d})\eta^2 + \frac{L\mu^4 d}{(d+2)^2}\kappa^2\eta\right]\left(\mathbb{E}f(p_t) - f^*\right)$$

$$- \left(\mathbb{E}f(p_{t+1}) - f^*\right) + (C_{\text{Ret}} + \frac{L}{2})\left(3B(\frac{2 + \mu^2\kappa^2}{d}) + 3\mathscr{E} + 3C_{\text{Ret}}^2\mu^2\right)\eta^2$$

$$+ \frac{\eta d}{2}\mathscr{F} + \frac{3}{4}d\eta\mu^2 C_{\text{Ret}}^2,$$

It has the same structure presented in [Lemma C.14](#), where we set

$$\theta_t = \frac{\eta}{6d}\|\nabla f(p_t)\|_{p_t}^2, \qquad \mathsf{S} = 6L(C_{\text{Ret}} + \frac{L}{2})(\frac{2 + \mu^2\kappa^2}{d})\eta^2 + \frac{L\mu^4 d}{(d+2)^2}\kappa^2\eta, \qquad \delta_t = \mathbb{E}f(p_t) - f^*,$$

$$G_t = (C_{\text{Ret}} + \frac{L}{2})\left(3B(\frac{2 + \mu^2\kappa^2}{d}) + 3\mathscr{E} + 3C_{\text{Ret}}^2\mu^2\right)\eta^2 + \frac{\eta d}{2}\mathscr{F} + \frac{3}{4}d\eta\mu^2 C_{\text{Ret}}^2.$$

Then we obtain

$$\min_{1 \le t \le T} \theta_t \le \frac{e^{\mathsf{S}T}}{T}\delta_1 + \max_{1 \le t \le T} G_t.$$

It leads to

$$\min_{1 \le t \le T} \|\nabla f(p_t)\|_{p_t}^2 \overset{(i)}{\le} \frac{6e^2[\mathbb{E}f(p_1) - f^*]}{\eta T/d} + \frac{6d}{\eta}\left[\frac{\eta d}{2}\mathscr{F} + \frac{3}{4}d\eta\mu^2 C_{\text{Ret}}^2\right]$$

$$+ \frac{6d}{\eta}\left[(C_{\text{Ret}} + \frac{L}{2})\left(3B(\frac{2 + \mu^2\kappa^2}{d}) + 3\mathscr{E} + 3C_{\text{Ret}}^2\mu^2\right)\eta^2\right].$$

where (i) selects

$$
\begin{cases}
\eta \le \sqrt{\dfrac{d}{T}} \sqrt{\dfrac{1}{18L(C_{\mathrm{Ret}} + \frac{L}{2})}} \\[2mm]
\mu^2 \le \min \left\{ \dfrac{1}{\kappa^2}, \sqrt{\dfrac{d}{T}} \dfrac{1}{18L^2(C_{\mathrm{Ret}} + \frac{L}{2})} \right\}
\end{cases}
\tag{21}
$$

such that $e^{T\mathsf{S}} \le e^2$, where $\frac{1}{\kappa^2}$ is considered as $+\infty$ when $\kappa = 0$. Given Equation (21), we further upper bound it as

$$
\min_{1 \le t \le T} \|\nabla f(p_t)\|_{p_t}^2
$$
$$
\le \frac{d}{\eta T} \left[ 6e^2 [\mathbb{E} f(p_1) - f^*] \right] + 3d^2 \mathscr{F} + \frac{9}{2} d^2 \mu^2 C_{\mathrm{Ret}}^2
\tag{22}
$$
$$
+ 6d\eta (C_{\mathrm{Ret}} + \frac{L}{2}) \left( 3B(\frac{2 + \mu^2 \kappa^2}{d}) + 3\mathscr{E} + 3C_{\mathrm{Ret}}^2 \mu^2 \right)
$$
$$
\overset{(i)}{\le} \frac{d}{\eta T} \left[ 6e^2 [\mathbb{E} f(p_1) - f^*] \right] + 3d^2 \left[ \frac{\mu^4 M_3^2}{9d^2(d+2)^2} + \frac{\mu^6}{144} M_4^2 + \frac{\mu^4}{(d+2)^2} \kappa^2 B \right] + \frac{9}{2} d^2 \mu^2 C_{\mathrm{Ret}}^2
$$
$$
+ 6d\eta (C_{\mathrm{Ret}} + \frac{L}{2}) \left( 3B(\frac{2 + \mu^2 \kappa^2}{d}) + 3\mu^2 \left[ \frac{4}{3} \frac{M_3^2}{d^3} + \frac{M_4^2 \mu^4}{288} \right] + 3C_{\mathrm{Ret}}^2 \mu^2 \right)
$$
$$
\le \frac{d}{\eta T} \left[ 54 [\mathbb{E} f(p_1) - f^*] \right] + \frac{M_3^2}{3} \frac{\mu^4}{d^2} + \frac{M_4^2}{48} \mu^6 + 3\mu^4 \kappa^2 B + 5d^2 \mu^2 C_{\mathrm{Ret}}^2
$$
$$
+ 54(C_{\mathrm{Ret}} + \frac{L}{2}) B\eta + 18(C_{\mathrm{Ret}} + \frac{L}{2}) \left[ \frac{4}{3} \frac{M_3^2}{d^3} + \frac{M_4^2 \mu^4}{288} \right] d\eta\mu^2 + 18(C_{\mathrm{Ret}} + \frac{L}{2}) C_{\mathrm{Ret}}^2 d\eta\mu^2
\tag{23}
$$

$$
= \mathcal{O}(\frac{d}{\eta T}) + \mathcal{O}(\eta) + \mathcal{O}(d^2 \mu^2),
$$

where (i) applies the formula of $\mathscr{E}$ and $\mathscr{F}$ given by Equation (7) and Equation (10), respectively. $\square$

## C.7 PROOF OF COROLLARY 2.10

We re-state this corollary to have a consistent notation as previous sections.

**Corollary C.19.** *Let $g$ be the Euclidean metric, and let $g'$ be a structure-preserving metric with respect to $g$. Under the same assumptions as Theorem 2.9, suppose that **either** of the following conditions holds:*

*(a) $g$ is geodesically complete; or*

*(b) the set of $\epsilon$-stationary points under $g$, $K := \{ p \in \mathcal{M} : \|\nabla_g f(p)\|_{p,g} \le \epsilon \}$, is compact.*

*Then it requires at most $T \le \mathcal{O}\left(\frac{d}{\epsilon^4}\right)$ iterations to achieve $\min_{1 \le t \le T} \mathbb{E}\left[\|\nabla f(p_t)\|_{p_t,g}^2\right] \le \epsilon^2$.*

*Proof.* For the item (a), we omit its proof as it is directly implied by setting $h \equiv 1$. Recall that we write $a \lesssim b$ if there exists a constant $\mathsf{C} > 0$ such that $a \le \mathsf{C} b$. Now we denote $g'_p(v, w) := h(p) g_p(v, w)$. Theorem C.18 implies that

$$
\min_{1 \le t \le T} \|\nabla f(p_t)\|_{p_t,g'}^2 \lesssim \sqrt{\frac{d}{T}}.
$$

It suffices to prove that if $p \in K$ is an $\epsilon$-stationary point under $g'$ then it must be an $\epsilon$-stationary point under $g$ (up to a constant scale). Note that

$$
\|\nabla_{g'} f(p)\|_{p,g'} = \frac{1}{\sqrt{h(p)}} \|\nabla_g f(p)\|_{p,g}.
$$

As the result, we obtain

$$\frac{1}{\max_{p \in \mathcal{M}} h(p)} \min_{1 \le t \le T} g_p(\nabla f(p), \nabla f(p)) \lesssim \sqrt{\frac{d}{T}}$$

$$\min_{1 \le t \le T} g_p(\nabla f(p), \nabla f(p)) \lesssim \max_{p \in \mathcal{M}} h(p) \sqrt{\frac{d}{T}}$$

We restrict two sides on the compact set (given by the condition (b))

$$K := \{p : \|\nabla_g f(p)\|_{p,g} \le \epsilon\}.$$

Because $h : \mathcal{M} \to \mathbb{R}$ is a continuous function, then it must be bounded over this compact set. Let this upper bound be $C$. Then we obtain (with absorbing $C$ into $\lesssim$)

$$\min_{1 \le t \le T} \|\nabla f(p)\|_{p,g}^2 \lesssim \sqrt{\frac{d}{T}}$$

By setting $\sqrt{\frac{d}{T}} \le \epsilon^2$, we obtain the complexity $T \gtrsim \frac{d}{\epsilon^4}$. $\qquad\square$

## D    EXPERIMENTAL DETAILS

In this section, we aim to include the omitted experimental details in Section 3.

**Hardware and System Environment**    We conducted our experiments on the personal laptop, equipped with AMD Ryzen 9 7940HS Mobile Processor (8-core/16-thread) and NVIDIA GeForce RTX 4070 Laptop GPU; however, GPUs are not required in our experiments. Our codes were tested using Python version 3.12.3. Additional dependencies are specified in the supplementary 'requirements.txt' file. All source codes attached.

### D.1    SYNTHETIC EXPERIMENT: IMPACT OF SAMPLING BIAS

**Construction of Quadratic Objective Functions**    We construct quadratic objective functions of the form $f_{\text{quadratic}}(x) = \frac{1}{2} x^\top (B + \xi) x$, where $B$ is a symmetric positive definite matrix that determines the landscape's curvature properties and $\xi$ is the data point independently sampled from $\mathcal{N}(0, 1)$ for each entry. The matrix $B$ is generated by first creating a random matrix $M \in \mathbb{R}^{d \times d}$ with entries drawn from a standard normal distribution $\mathcal{N}(0, 1)$, then forming $B = M^\top M + d I_d$ to ensure positive definiteness with a regularization term $d I_d$ that controls the minimum eigenvalue.

**Construction of Logistic Objective Functions**    For logistic objective functions, we construct the empirical risk minimization problem $f_{\text{logistic}}(x) = \frac{1}{n} \sum_{i=1}^{n} \log(1 + \exp(-y_i \zeta_i^\top x)) + \frac{\lambda}{2} x^\top B x$, where $B$ is generated as the same way as the quadratic function and $\{(\zeta_i, y_i)\}_{i=1}^{n}$ represents the training dataset with feature vectors $a_i \in \mathbb{R}^d$ and binary labels $y_i \in \{-1, +1\}$. The feature matrix $X = [x_1, \ldots, x_n]^\top \in \mathbb{R}^{n \times d}$ is generated from a standard normal distribution $\mathcal{N}(0, 1)$. A ground truth weight vector $w^* \in \mathbb{R}^d$ is generated from $\mathcal{N}(0, 1)$ and then normalized to unit length. The binary labels $y_i \in \{-1, +1\}$ are generated by first computing logits $x_i^\top w^*$, then converting to probabilities $p_i = 1/(1 + \exp(-x_i^\top w^*))$, and finally sampling $y_i$ according to Bernoulli$(p_i)$ before converting to the $\{-1, +1\}$ encoding. The regularization parameter $\lambda$ is chosen as $\lambda = 0.1$.

**Construction of Riemannian Metric $g_A$**    We design a Riemannian metric on the ambient Euclidean space by defining a symmetric positive definite matrix $\mathbf{A}$ with extreme conditioning properties. Specifically, the metric tensor is constructed by generating a random orthonormal matrix $\mathbf{Q}$ via QR decomposition, prescribing eigenvalues that span geometrically from $\lambda_{\min} = 1$ to $\lambda_{\max} = 10^4 \lambda_{\min}$, and forming $\mathbf{A} = \mathbf{Q} \mathbf{\Lambda} \mathbf{Q}^\top$, where $\mathbf{\Lambda}$ is the diagonal matrix of these eigenvalues. This construction yields a highly anisotropic Riemannian manifold with a condition number of $\mathbf{A}$ equal to $10^4$, creating challenging geometric landscapes for optimization algorithms. The resulting metric induces Riemannian gradients of the form $\mathbf{A}^{-1} \nabla f(\mathbf{x})$, fundamentally altering the optimization dynamics compared to standard Euclidean methods.

**Hyper-Parameters**   Each method uses 16 random directions per iteration with a perturbation stepsize $\mu = 10^{-4}$ for gradient estimation. The algorithms were run for 500,000 iterations with learning rates of $10^{-3}$ (quadratic) and $10^{-5}$ (logistic), and results were averaged over 16 independent runs to ensure statistical reliability. All curves are smoothed using a moving average with a window size of 5,000 iterations, and confidence bands represent 10th–90th percentiles across runs to visualize convergence variability.

## D.2   Synthetic Experiment: MSE vs. Curvature

**Riemannian Metric Construction**   We work on the $d$-dimensional probability simplex

$$\Delta^d := \{p \in \mathbb{R}^{d+1} \mid \sum_{i=1}^{d+1} p_i = 1, 0 < p_i < 1\},$$

and endow its interior (identified with the first $d$ coordinates) with a structure-preserving Riemannian metric (Definition 2.5) conformally equivalent to the canonical Euclidean metric $g^E$:

$$\tilde{g}^{(\beta)} = e^{2\phi_\beta(p)} g^E,$$

where the conformal factor is

$$\phi_\beta(p) \; = \; \tfrac{1}{2} \beta \log h(p), \qquad h(p) \; = \; 1 + \sum_{i=1}^{d+1} \frac{1}{p_i^2} - \frac{1}{d+1} \Big( \sum_{i=1}^{d+1} \frac{1}{p_i} \Big)^2.$$

Varying the scalar $\beta > 0$ sharpens or flattens the metric. We examine four choices $\beta \in \{0.5, 1.0, 1.5, 2.0\}$. At the fixed reference point $p_0 \in \Delta^d$ (drawn once from the Dirichlet distribution and held constant throughout the experiment) we measure the mean-squared error of a symmetric zeroth-order gradient estimator (Equation (3)) with using the first-order approximation of the exponential map as the retraction, where the perturbation stepsize $\mu$ is set to 0.1. We note that under this approximation, the retraction degenerates to the naive Euclidean perturbation; we note that we are using a fixed point $p_0$, it doesn't trigger the out-of-domain issue of the incomplete Riemannian manifolds when $\mu$ is appropriately selected. The MSE is evaluated using the corresponding structure-preserving metric instead of the original Euclidean metric; the conformal scaling $h(p)^\beta$ is applied consistently both when sampling directions ($\|v\|_{\tilde{g}} = 1$) and when converting the Euclidean gradients of the test functions (quadratic and Kullback–Leibler distance to the uniform distribution) into true Riemannian gradients.

**Sectional Curvature Evaluation**   Instead of using $\beta$ as the x-axis, we compute the sectional curvature $K_{\tilde{g}^{(\beta)}}(p_0)$ of each metric at $p_0$ to reflect the true relation between the intrinsic curvature and the estimation error. Let $\phi = \phi_\beta$; then $\tilde{g}^{(\beta)} = e^{2\phi} g^E$ is a warped Euclidean metric whose curvature depends solely on $\phi$. We draw an orthonormal pair $(v, w)$ in the Euclidean tangent space $T_{p_0}\Delta^d$ via Gram-Schmidt method, rescale them so that $\|v\|_{\tilde{g}} = \|w\|_{\tilde{g}} = 1$, and evaluate

$$K(p_0; v, w) = e^{-2\phi(p_0)} \Big( \|\nabla\phi(p_0)\|_2^2 - \langle \mathrm{Hess}\, \phi(p_0)\, v, v \rangle - \langle \mathrm{Hess}\, \phi(p_0)\, w, w \rangle$$
$$- \langle \nabla\phi(p_0), v \rangle^2 - \langle \nabla\phi(p_0), w \rangle^2 \Big),$$

where gradients and Hessians are taken with respect to the ambient Euclidean coordinates. Because $\tilde{g}^{(\beta)}$ is *isotropic* up to the conformal factor, a single random 2-plane suffices; the resulting scalar is recorded as $K(p_0)$ for that $\beta$. These four curvature values, monotonically decreasing as $\beta$ grows, serve as the horizontal tick labels in Figure 1b.

**Hyper-Parameters**   For each metric, we run 50,000 independent zeroth-order gradient trials, each trial drawing one random Riemannian unit direction and applying Equation (3) to estimate the gradient with using the perturbation stepsize $\mu = 0.1$ and using the exponential map as the retraction. The reference point $p_0 \in \Delta^4$ is sampled once and held fixed, so that changes in estimator accuracy stem solely from the chosen metric. Closed-form gradients are available for both test functions, Euclidean and KL distance to the uniform distribution. We record the mean-squared error $h(p_0)\|\widehat{\nabla}f - \nabla f\|^2$ for each trial. The resulting 50,000 errors per setting are summarized with log-scale box plots whose boxes span the inter-quartile range and whiskers cover the 10th–90th percentiles (outliers omitted).

Table 1: Hyper-parameter settings for gradient-based mesh optimization experiments.

| Fixed Hyperparameter | Symbol | Value |
|---|---|---|
| Fine-mesh size | $M_{\text{fine}}$ | $200 \times 200$ |
| Number of nodes in the fine mesh | $N_{\text{fine}}$ | 40,000 |
| Coarse-mesh size | $M_{\text{coarse}}$ | $20 \times 20$ |
| Number of nodes in the coarse mesh | $N_{\text{coarse}}$ | 400 |
| Sampled nodes in each iteration | - | 120 |
| All nodes positions | $P$ | - |
| Total iterations | $T$ | 20,000 |
| **Tunable Hyperparameter** | | |
| Random directions | - | 4 |
| Perturbation stepsize | $\mu$ | $10^{-1}$ |
| Learning rate | $\eta$ | $\{300, 400, 500\}$ |

### D.3 GRADIENT-BASED MESH OPTIMIZATION

In our work, we consider the black-box mesh optimization problem. In the well-known CFD-GCN model (Belbute-Peres et al., 2020), additional efforts are taken to allow the position of nodes to support the auto-differentiation in the SU2 PDE solver; however, in most of existing finite-volume numerical solvers, the positions of mesh nodes are typically not differentiable. Therefore, we need to apply the zeroth-order optimization approach.

**Construction of Mesh Objective Function** Let $P = \{p_i\}_{i=1}^N \subset \mathbb{R}^2$ be interior node positions of the given mesh with boundary nodes fixed. Given $P$, the coarse mesh induced by $P$ defines a PDE state $\hat{u}_P$ (solved on $P$). Then we interpolate it into the fine mesh $M_{\text{fine}}$ to obtain the PDE state $u_P$. The objective is the mean-squared error (MSE) to a fixed fine-grid reference $u_{\text{ref}}$:

$$f_{\text{mesh}}(P) = \frac{1}{N_{\text{fine}}} \big\| u_P - u_{\text{ref}} \big\|_2^2,$$

where $N_{\text{fine}}$ denotes the number of nodes in the fine mesh. The randomness in this objective comes from the random sampling over the nodes; instead of taking all nodes to be updated, each step we will only sample a part of nodes to be updated. In our experiments, we set the size of coarse mesh to be $20 \times 20$ and the size of fine mesh to to be $200 \times 200$. Each time, we will randomly sample $30\% \times 20 \times 20 = 120$ nodes to update.

**Construction of Mesh Parameterization** Each interior node is updated in **barycentric coordinates** $b \in \Delta^{m-1}$ with respect to its incident cell (with vertices $\{v_j\}_{j=1}^m$), i.e., $p(b) = \sum_{j=1}^m b_j v_j$. This coordinate guarantees feasibility ($b_j > 0$, $\sum_j b_j = 1$), which naturally results in a probability simplex structure. Under the canonical inclusion embedding, this manifold is geodesically incomplete and hence feasible for our proposed approach.

**Construction of Structure-Preserving Metric** We endow $\Delta^{m-1}$ with the structure-preserving conformal metric $\tilde{g}^{(\beta)}$ as defined in Appendix D.2, and use the first-order approximation of the exponential map of $\tilde{g}^{(\beta)}$ as the retraction (Definition B.3). We note that this approximation requires to set the length of perturbation vectors to be sufficiently small to ensure the accuracy of the retraction; this requirement can be satisfied by adopting the same technique as the soft projection trick used in Figure 5a. We always assume this requirement is satisfied throughout the training.

**Hyper-Parameters** Each iteration uses 4 random directions with perturbation stepsize $\mu = 10^{-1}$. Optimization runs for $T = 20,000$ iterations with learning rate $\eta \in \{300, 400, 500\}$ (we report the best curve among these hyper-parameters). All curves are smoothed with a moving-average window of 2,000 iterations. For all other other estimator-dependent hyper-parameters, we have included all of them in the configuration files along with source codes.

### D.4 Additional Discussions on Stability

To further evaluate the robustness of the proposed method, we conducted repeated experiments using 5 independent random seeds. The results are illustrated in Figure 7. We note that both our method and the *Reversion* method exhibit low variance and consistently show the narrowest error bands (smallest shaded areas), indicating that they are sufficiently stable. In contrast, the naive *Unconstrained* method suffers from the highest variance.

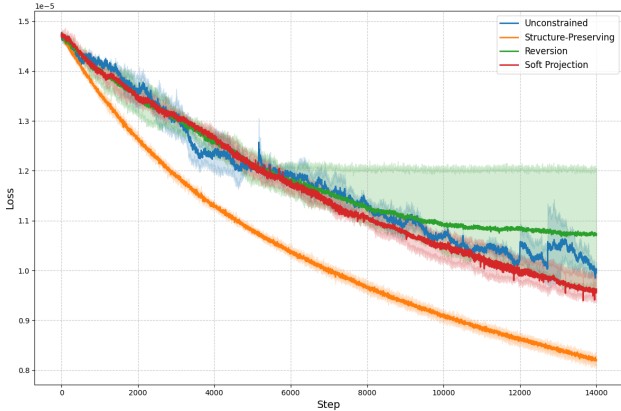

Figure 7: Training loss trajectories averaged over 5 independent random seeds. The solid lines represent the mean loss, while the shaded areas indicate the min-max deviation. Our structure-preserving method demonstrates superior stability and consistently lower loss.

## E Limitations and Future Work

Our work also presents several limitations which potentially point to future research directions. A potential limitation is the rejection sampling's reliance on the eigen-decomposition of the metric matrix, which presents a significant computational bottleneck when applied to high-dimensional problems. Developing more scalable algorithms (e.g. Randomized SVD or iterative solvers) to efficiently handle high-dimensional manifolds is therefore an important future direction. Additionally, while our theory guarantees the existence of a structure-preserving metric (Theorem 2.6), its practical construction currently relies on a case-by-case design and lacks a general construction. Moreover, our construction is based on the conformal transformation; a valuable direction for future work is to explore whether a broader class of structure-preserving transformations exists beyond the current scaling and to develop more general constructive methods that are better compatible for zeroth-order optimization.

## F The Use of Large Language Models (LLMs)

In preparing this manuscript, we employed Large Language Models (LLMs) as general-purpose assistive tools in the following ways:

- *Literature review support.* We used the Deep Research functionality provided by existing AI platforms to help gather references and draft preliminary summaries of related work.
- *Language refinement.* We used AI chatbots hosted on multiple platforms to generate the abstract and to improve the clarity, style, and readability of the manuscript.
- *Proof verification.* We used AI chatbots to check the logical consistency, correctness, and completeness of our formal proofs.
- *Codes Generation.* We also applied the AI agent to generate a part of experimental codes.

All LLM-assisted outputs were critically reviewed, verified, and, where necessary, revised by the authors. We take full responsibility for the content of this manuscript. LLMs were not involved in generating research ideas, drawing scientific conclusions, or contributing original insights.

Table 2: A summary of notations in Riemannian manifolds.

| Notations | Definition |
|---|---|
| Smooth Manifold ($\mathcal{M}$) | A $d$-dimensional second-countable Hausdorff topological space where each point $p$ has a neighborhood $U_p$ diffeomorphic to $\mathbb{R}^d$. |
| Deviation ($v$) | A linear mapping $v : C^\infty(U_p) \to \mathbb{R}$ satisfying the product rule: $v(fg) = v(f) \cdot g(p) + v(g) \cdot f(p)$. |
| Tangent Space ($T_p\mathcal{M}$) | The real vector space of all deviations at a point $p \in \mathcal{M}$. |
| Cotangent Space ($T_p^*\mathcal{M}$) | The dual space of the tangent space $T_p\mathcal{M}$; the space of all linear maps $\psi : T_p\mathcal{M} \to \mathbb{R}$. |
| Tangent Bundle ($T\mathcal{M}$) | The disjoint union of all tangent spaces: $T\mathcal{M} := \{(p,v) \mid p \in \mathcal{M}, v \in T_p\mathcal{M}\}$. |
| Immersion | A smooth map $f : \mathcal{M} \to \mathbb{R}^n$ whose differential $df\|_p$ is injective at every $p \in \mathcal{M}$. |
| Embedding | An immersion that is also a homeomorphism onto its image $f(\mathcal{M})$. |
| Vector Field ($X$) | A smooth map (section) $X : \mathcal{M} \to T\mathcal{M}$ such that $X(p) \in T_p\mathcal{M}$ for all $p \in \mathcal{M}$. |
| $\mathfrak{X}(\mathcal{M})$ | The space of all vector fields on the manifold $\mathcal{M}$. |
| Riemannian Metric ($g$) | A smooth assignment of an inner product $g_p : T_p\mathcal{M} \times T_p\mathcal{M} \to \mathbb{R}$ to each tangent space $T_p\mathcal{M}$. Also denoted $\langle \cdot, \cdot \rangle_p$. |
| Riemannian Manifold (($\mathcal{M}, g$)) | A smooth manifold $\mathcal{M}$ equipped with a Riemannian metric $g$. |
| $n$-Euclidean Metric | A metric $g$ induced by a smooth embedding $\phi : \mathcal{M} \to \mathbb{R}^n$ via the pullback $g_p^E(v, u) = \langle d\phi\|_p(v), d\phi\|_p(u) \rangle$. |
| Levi-Civita Connection | The unique affine connection on $\mathfrak{X}(\mathcal{M})$ that is torsion-free and metric-compatible. |
| Geodesic ($\gamma$) | A smooth curve $\gamma : I \to \mathcal{M}$ whose velocity vector $\gamma'(t)$ satisfies the geodesic equation $\nabla_{\gamma'(t)}\gamma'(t) = 0$. |
| Exponential Map ($\exp_p$) | A map $\exp_p : T_p\mathcal{M} \to \mathcal{M}$ defined by $\exp_p(v) := \gamma(1)$, where $\gamma$ is the unique geodesic with $\gamma(0) = p$ and $\gamma'(0) = v$. |
| Retraction (Ret) | A smooth map $\mathrm{Ret} : T\mathcal{M} \to \mathcal{M}$ satisfying $\mathrm{Ret}_p(0) = p$ and $d\mathrm{Ret}_p\|_0 = \mathrm{id}_{T_p\mathcal{M}}$. It approximates the exponential map. |
| Gradient ($\nabla f$) | The vector field $\nabla f(p) := (df\|_p)^\sharp$, where $\sharp$ is the musical isomorphism $T_p^*\mathcal{M} \to T_p\mathcal{M}$ induced by the metric $g$. |
| Riemannian Stochastic Optimization Problem | $\min_{p \in \mathcal{M}} f(p) = \mathbb{E}_{\xi \sim \Xi}[f(p; \xi)]$, where $f(\cdot; \xi) : \mathcal{M} \to \mathbb{R}$ is a smooth function relying on $\xi$ drawn from the data distribution $\Xi$. |
| Symmetric zeroth-order estimator | $\widehat{\nabla} f(p) = \frac{f(\mathrm{Ret}_p(\mu v)) - f(\mathrm{Ret}_p(-\mu v))}{2\mu} v$, where $\mu$ is the perturbation stepsize and $v$ is uniformly sampled from the unit ball in $T_p\mathcal{M}$. |
| SGD update rule | $p_{t+1} = \mathrm{Ret}_{p_t}\big(\eta \widehat{\nabla} f(p_t; \xi_t)\big)$, where $\eta$ is the learning rate. |

