# OpenReview forum: "Riemannian Zeroth-Order Gradient Estimation with Structure-Preserving Metrics for Geodesically Incomplete Manifolds"
_ICLR.cc/2026/Conference — ICLR 2026 Poster_

### Official Review · Reviewer_VPTy · 2025-10-26

**Soundness:** 2
**Presentation:** 2
**Contribution:** 2
**Rating:** 2
**Confidence:** 3

**Summary:**

The authors consider the minimization of a real-valued smooth function on a manifold, using only function evaluations. Contributions are as follows:
- To point an issue with existing zero-th order optimization methods, as follows. The exponential mapping is often defined on an open subset of the tangent space that contains the zero vector. There exist algorithms that rely on the exponentail mapping, but do not account for this fact. Unless the manifold is proved to be geodesically complete, that is the exponential mapping is defined for any tangent vector, such algorithms are ill-defined (theoretical guarantees are not applicable, possible breakdown in practice).
- A theoretical guarantee that the metric of a geodesically incomplete manifold may modified such that it becomes geodesically complete, while preserving the stationary points of any function to be minimized (Th. 2.2).
- a zero-th order gradient estimator that relies on two function evaluations, two geodesic computation, and a sampling routine on the tangent space. A mean-squared error analysis is provided (Th.  2.3).
- an  analysis of the complexity of SGD with zero-th order gradient estimator for finding $\epsilon$-stationary points (Th. 2.5),  and an analysis of the complexity of SGD and mapping from one metric to a geodesically complete metric, and back to the original metric (Coro 2.6).

**Strengths:**

- the paper identifies a reasonable issue in existing works,
- the result Th. 2.3 account for manifolds curvature, a difficult topic, in a reasonably clear way,
- one experiment (sec. 3.3) stems from a real-world application.

**Weaknesses:**

The following aspects motivate my current assessment of the paper.
- The scheme of (eq. 2, $f(Exp(\mu v) - f(Exp(-\mu v)))$) differs from the two references (l. 46 -  $f(Exp(\mu v) - f(x))$). Yet, this difference is not acknowledged, motivated, nor discussed.
- I fail to see why does working with a non-euclidean metric precludes the use of previous estimators, such as in Li et al, 2023b.
- The work focuses on situations where the manifold and its metric are not geodesically complete. However, the introduction does not provide any practical situations where this situation occurs. The only example in the paper is optimization on the simplex (fig. 1, and experiments), or a union thereof. I am not convinced this example alone is sufficient to generate interest from the iclr optimization community.
- l. 198-202: I fail to see why existing results in Riemannian optimization may not be applied with arbitrary metrics, different from the ones of the ambiant space.
- The only example with relevant application of a manifold that is not geodesically complete is the unit simplex (Fig 1), or a collection thereof (numerical experiments).
- It is not clear to me that the proposed procedure to sample uniformly on an ellipse is novel (a contribution of this work) or classical. I am not an expert of distribution sampling, but I suspect the rejection sampling on an ellipse is classical.
- The same notation, $\hat{\nabla} f(p)$, is used for the gradient estimator when sampling with exponential mapping and retraction (eq. 2 and 5). It is thus not clear to which estimator Theorems 2.3 and 2.5 apply.
- As far as I understand, while Th. 2.2 guarantees existence of a structure preserving metric, this proof does not help in designing the structure preserving metric in a practical situation, as for instance the example of fig 1 or the experiments. The applicability of the method thus fully relies on the practitionner.
- Experiments, sec 3.1: authors compare SGD with rescale sampling and rejection sampling in terms of SGD iterations. This does not account for the additional complexity of the rejection sampling, relative to the rescale sampling. As such, the plots does not inform on the performance of the complete method (SGD + gradient sampling scheme).
- Experiments, sec 3.3: authors compare variants of SGD relative to the number of iterations, with four different strategies for the update. Besides, there is one trajectory for each method, which does not provide any information on the variance of the methods.  Again, I believe this does not inform on the performance of the overall methods, which I believe is ulitmately the metric of interest.
- Appendix C (proofs) shows several serious issues:
  - Lemma C.5: in view of $f*$ definition, there holds, by definition, $B = 0$.
  - Lemma C.5, l. 1005-1008: the proof is not detailled enough to be checked.
  - the lemmas C.9, C.10 are not referenced anywhere. Lemmas C.9, C.10, and C.13 are not used anywhere as far as I can see.
  - l. 1072: introduces $g$\# but does not uses it.
  - l. 1106: $n$ is used but not defined there, nor in the referenced Lemma C.8.
  - l. 1107: the Ricci tensor is nowhere defined, nor discussed except in this lemma. This lemma is not used is the appendix as far as I can see.

Minor points
- l. 42: what is nondifferentiable modules?
- l. 90: writing issue "while maintain"
- l. 127: syntax issue "present main"
- l. 134: syntax issue "we establishes"
- l.155-160 : ($\epsilon$)stationary point not defined
- l. 255: syntax issue in "metric $g$ the choice"
- l. 303: I would appreciate that the theorems assumptions are stated in the main body rather than supplementary
- l. 305: the SGD dynamics do not solve (1) (i.e. converge to the global minimizers of $f$ on M)
- l. 372: $g_A$ not defined
- l. 408: syntax issue "aligns our"
- l. 939, last sentence of section: syntax issue.
- l. 1015, lemma C.6: syntex "there exists a function [...] is proper"

**Questions:**

- Why does the scheme of (eq. 2, $f(Exp(\mu v) - f(Exp(-\mu v)))$) differs from the two references (l. 46 -  $f(Exp(\mu v) - f(x))$)?
- Why does working with a non-euclidean metric precludes the use of previous estimators, such as in Li et al, 2023b?
- What are situations where the manifold and its metric are not geodesically complete?
- l. 198-202: why do existing results in Riemannian optimization may not be applied with arbitrary metrics, different from the ones of the ambiant space?
- What are examples with relevant applications of manifold that are not geodesically complete?
- The same notation, $\hat{\nabla} f(p)$, is used for the gradient estimator when sampling with exponential mapping and retraction (eq. 2 and 5). To which estimator do Theorems 2.3 and 2.5 apply.
- As far as I understand, while Th. 2.2 guarantees existence of a structure preserving metric, this proof does not help in designing the structure preserving metric in a practical situation, as for instance the example of fig 1 or the experiments. Am I correct in this? The applicability of the method thus fully relies on the practitionner. Can this be addressed by your theory?
- Experiments, sec 3.1 and 3.3. Can you report experiments relative to time, and with variance indicators for 3.3?
- Appendix C (proofs):
  - Lemma C.5, can you review the proof with more details?
  - What is the use of lemmas C.9, C.10, C.13?
  - l. 1072: why introduce $g$\# but not use it?
  - l. 1106: what is $n$?
  - l. 1107: the Ricci tensor is nowhere defined, nor discussed except in this lemma. This lemma is not used is the appendix as far as I can see. Why is that?

---

> ### Author Response · Authors · 2025-11-18
> **Rebuttal Comment (1/4)**
>
> We sincerely appreciate the reviewer’s careful reading and comprehensive comments on each issue. Below, we provide our point-by-point responses to each of the concerns raised.
>
> ---
>
> > **W1** & **W2** & **W4** (Existing Zeroth-Order Estimators)
> >
> > **Q1:** What does the scheme of eq.2 $f(\exp_p(\mu v))-f(\exp_p(-\mu v))$ differs from the two references (I.46 $f(\exp_p(\mu v))-f(p)$)?
> >
> > **Q2:** Why does working with a non-Euclidean metric precludes the use of previous estimators, such as in Li et al, 2023b?
> >
> > **Q4:** l. 198-202: why do existing results in Riemannian optimization may not be applied with arbitrary metrics, different from the ones of the ambient space?
>
> * **Response:** We appreciate the opportunity to clarify this point. To be clear, we are not proposing a new estimator. The scheme used in our paper (Eq. 2) is not fundamentally different from those in existing literature, and either our two-sided version (Eq. 2) or the one-sided version (used in the reference) can be used.
>
>     That is, working with a non-Euclidean metric doesn't precludes the use of previous estimators; **however**, it does preclude two critical components in prior zeroth-order approaches: (i) the projection technique used to ensure the estimator is well-defined, and (ii) their corresponding convergence analyses, which rely on those projections.
>
>     * *(i) Preclusion of Existing Projection Techniques*
>
>         Existing literature (on Riemannian zeroth-order gradient estimation) requires the geodesic completeness; that is, the exponential mapping $\exp_p$ must be well-defined at the point $\mu v \in T_p \mathcal{M}$. To achieve this, it treats the manifold $\mathcal{M}$ as a sub-manifold of an Euclidean space (called the *ambient space*). When  $\exp_p$ is not well-defined (e.g. $\exp_p(\mu v)$ falls outside $\mathcal{M}$), they use a projection operator $P$ to project the point  $\exp_p(\mu v)$ back onto the manifold $\mathcal{M}$.
>
>         This projection-based approach is fundamentally incompatible with a general Riemannian metric, as it is difficult to know how the manifold is embedded into a Euclidean space. As a result, we cannot construct the required projection operator.
>
>     * *(ii) Preclusion of Existing Convergence Analyses*
>
>         Our paper resolves the geodesic completeness issue by using a structure-preserving metric. Under this metric, it also allows us to directly apply standard estimator schemes  (including Eq.2 $f(\exp_p(\mu v))-f(\exp_p(-\mu v))$ and the one-side version $f(\exp_p(\mu v))-f(p)$ from  Li et al, 2023b). However, existing convergence analyses in the literature crucially rely on embedding the manifold into an Euclidean space, an approach that is not applicable for a general Riemannian metric.
>
>         Consequently, we must develop a purely intrinsic convergence analysis that does not depend on any embedding. We further emphasize that this intrinsic perspective reveals a novel connection between the manifold’s curvature and the mean-squared error of the zeroth-order estimator (Theorem 2.3).

---

> > ### Author Response · Authors · 2025-11-18
> > **Rebuttal Comment (2/4)**
> >
> > > **W3** & **W5** & **W8** & **W9** (Practical situations)
> > >
> > > **Q3:** What are situations where the manifold and its metric are not geodesically complete?
> > >
> > > **Q5:** What are examples with relevant applications of manifold that are not geodesically complete?
> >
> > * **Response:** From a mathematical perspective, we have two widely observed scenarios where a Riemannian manifold is geodesically incomplete:
> >
> >     * **Removing a Closed Subset from a Complete Manifold**
> >
> >         If we remove a closed subset $C$ from a complete Riemannian manifold $M$, the remaining set $M \setminus C$ (endowed with the inherited Riemannian metric) is a manifold that is generally no longer geodesically complete. This is because a geodesic in the original complete manifold $M$ might have passed through the removed closed subset $C$.
> >
> >         This setting can be applied to design a new version of a matrix optimizer. For example, consider the space of all $m \times n$ matrices, which is a complete Riemannian manifold (with the Riemannian metric induced by the Frobenius matrix norm). However, if we require one specific element to be non-zero (e.g. to propose a new optimizer), this constraint removes the closed set $\{ M : M_{i,j}=0 \}$. The resulting manifold is no longer geodesically complete.
> >
> >     * **A Riemannian manifold as an Open Set of a Euclidean Space**
> >
> >         This setting is a very common and specific instance of the first scenario. It includes many straightforward examples: the open interval, the probability simplex, and the open unit disk. When endowed with the canonical Euclidean metric, they are naturally geodesically incomplete Riemannian manifolds, as geodesics (which are straight lines) will exit these sets in finite time.
> >
> >         This is a fundamental challenge in constrained optimization. When we need to conduct optimization over these open subsets, algorithms (like gradient descent or Riemannian gradient methods) may propose steps that "leave" the feasible set.
> >
> >     From the practical side, we provide a few real-world examples demonstrating the necessity of developing the Riemannian zeroth-order optimization framework for those that are not geodesically complete (these applications will also be included in the revision).
> >
> >     * **Mesh Optimization**: In physical simulations, mesh optimization is essential for improving discretized surface quality. Modern neural physical models, such as CFD-GCN, adjust vertex positions by optimizing a quality metric, usually involving an external PDE solver. A major bottleneck is the requirement to implement auto-differentiation through this solver to obtain gradients, which is fundamentally difficult.  Riemannian zeroth-order optimization offers a compelling alternative by avoiding this gradient calculation. In this setting, the manifold consists of the valid configuration space of vertex positions. This manifold, however, is geodesically incomplete under the Euclidean metric, because configurations on the boundary (e.g., a vertex on an edge) are excluded to prevent numerical instability.
> >
> >     * **Irrigation System Layout Design**: This application seeks to optimize the physical coordinates of sprinklers to maximize water coverage. The coverage objective function is often a complex, non-differentiable simulation (e.g., modeling spray overlap, pressure, and wind), making it difficult to compute gradients. Riemannian zeroth-order optimization provides a gradient-free solution. The underlying manifold is the configuration space of valid sprinkler positions, defined by the open set within the field's boundaries. This manifold is geodesically incomplete, as typically we cannot directly put the sprinklers on the boundary of the field.
> >
> >     * **Covariance Matrix Estimation**: This is a fundamental problem in multivariate statistics and machine learning, essential for tasks like PCA and Gaussian modeling. The goal is to find a matrix that best represents the data's covariance, often by minimizing a loss function (e.g., maximizing likelihood). The underlying manifold is the set of all $d\times d$ \textit{positive definite matrices}, denoted $S_d^{++}$. A matrix $C$ is in this manifold if it is symmetric and $x^T C x > 0$ for all non-zero vectors $x \in \mathbb{R}^d$. This manifold is geodesically incomplete because it is an open convex cone.

---

> ### Author Response · Authors · 2025-11-18
> **Rebuttal Comment (3/4)**
>
> > **W7** (Notations - exponential mapping and retractions)
> >
> > **Q6:** The same notation, $\hat{\nabla} f(p)$, is used for the gradient estimator when sampling with exponential mapping and retraction (eq. 2 and 5). To which estimator do Theorems 2.3 and 2.5 apply.
>
> * **Response:** We appreciate the reviewer’s careful reading. In the revision, we have clarified exactly which estimator is used in each result. Specifically, in Theorem 2.3, we now explicitly state that the estimator is constructed using the exponential map. In Theorem 2.5, we clarify that the estimator is constructed using the retraction, as defined in Eq. (4).
>
> > Q7: As far as I understand, while Th. 2.2 guarantees existence of a structure preserving metric, this proof does not help in designing the structure preserving metric in a practical situation, as for instance the example of fig 1 or the experiments. Am I correct in this? The applicability of the method thus fully relies on the practitionner. Can this be addressed by your theory?
>
> * **Response:** We appreciate the reviewer's insightful question regarding the construction of the structure-preserving metric. The reviewer is correct that Theorem 2.2 itself only guarantees existence. While we do provide a specific construction for this metric in *Theorem C.17 (Appendix C.3, Line 1488)*, it inherently relies on the non-constructive proof of a smooth proper function (Lemma C.6).  This is why a case-by-case construction is necessary. For each specific manifold, one may explicitly construct a suitable function that is both proper and smooth then apply the construction in Theorem C.17.
>
>     We also highlight that the specific construction is not that hard. The general idea behind this construction is to "stretch" the distance when a point $p$ moves towards the boundary of the domain of exponential map. As shown in our Figure 1, in the standard metric, if the red point $p$ moves upward for three steps along the geodesic, it will be out of the simplex. In the structure-preserving metric, we re-scale the distance on the manifold to make three steps along the geodesic not as far as before, thus ensuring the point remains within the simplex.
>
> > Appendix C (proofs): Lemma C.5 details.
>
> * **Response:** We appreciate the reviewer's suggestion. In response, we have revised Lemma C.5 to improve its readability and added omitted steps. We have also included the missing statement needed to establish that $B\geq 0$.
>
>     A minor clarification: as $B$ is defined as $f^\star - \mathbb{E}\_{\xi \sim \Xi} f_\xi^\star$, where $f^\star$ is the minimal value of the expected loss, and $ \mathbb{E}\_{\xi \sim \Xi} f_\xi^\star$ is the expectation of the minimal value of each individual loss, it typically cannot be zero.
>
> > Appendix C (proofs): What is the use of lemmas C.9, C.10, C.13?
> >
> > Lemma C.7: l. 1072: why introduce $g^\sharp$ but not use it?
> >
> > Lemma C.9: l. 1106: What is $n$?
>
> * **Response:**  We appreciate the reviewer’s questions and the opportunity to clarify these points.
>
>     * **Lemma C.13:** The purpose of Lemma C.13 is to bridge our Assumption C.2 with the commonly used uniformly bounded gradient assumption. Specifically, the uniformly bounded gradient assumption automatically implies Assumption C.2 by Lemma C.13 (see the remark discussion of Assumption C.2). This demonstrates that our assumption is strictly weaker than the uniformly bounded gradient assumption.
>
>     * **Lemma C.10:** This lemma is used implicitly in Appendix, Eq.(9), where it is applied to upper bound the Ricci curvature in terms of the sectional curvature. We have revised the proof to add an explicit reference and clarify the notation to make this usage clear.
>
>     * **Lemma C.7 and Lemma C.9:** We also deeply thank the reviewer for bringing it to our attention: Lemma C.7 and Lemma C.9 are not used in our manuscript. The $g^\sharp$ and $n$ notations were introduced in these lemmas, which were also unused and removed in our revision.
>
>         (Currently, we color them gray in our revision, indicating that they will be removed later. As a related note, Lemma C.8, which was previously used to prove the now-removed Lemma C.9, is still required for Theorem C.18. We have updated the proof of Theorem C.18 to explicitly reference it.)
>
> > Lemma C.10: l. 1107: the Ricci tensor is nowhere defined, nor discussed except in this lemma. This lemma is not used is the appendix as far as I can see. Why is that?
>
> * **Response:** We appreciate this careful observation. Regarding the term: The reviewer is correct that "Ricci tensor" was imprecise. We have revised this to "Ricci curvature" for consistency. This is the same curvature used in  Eq. (17). We have updated the manuscript to add an explicit reference to this lemma where it is applied to resolve any ambiguity.

---

> > ### Author Response · Authors · 2025-11-18
> > **Rebuttal Comment (4/4)**
> >
> > #### Minor Points
> >
> > We thank the reviewer for the detailed suggestions. We have incorporated all of them, and the corresponding revisions are marked in red in the updated manuscript.
> >
> > A few clarifications:
> >
> > * (l. 42 what is nondifferentiable modules?): In many machine learning applications, the neural network is cooperated with an external program; e.g. a numerical PDE solver. The role of the neural network is to correct the error in this external program. Non-differentiable modules refer to those external software which doesn't support the auto-differentiation.
> > * (l. 303 move assumptions to the main body): We appreciate this suggestion. We have moved all assumptions to the main body in the revision.
> >
> > ---
> >
> > Lastly, we again greatly appreciate the constructive feedback and believe these edits will meaningfully strengthen the paper. We hope the improved clarity and completeness address the reviewer’s concerns.
> >
> > The Authors

---

> > > ### Comment · Reviewer_VPTy · 2025-11-26
> > >
> > > I deeply thank the authors for the extensive comments, explanations, and updates to the paper.
> > >
> > > With these updates, I find the positioning relative to the literature and practical situations convincing, the math easier to understand, and the appendix improved.
> > >
> > > About Appendix C, Lemma C.5 (updated version):
> > > 1. I am still puzzled by the fact that, quoting your text, $B = 2L[f* - {E}_{\xi} f_{\xi}*]$ (l. 1063), $f* = {E}_\xi [f_xi*]$ (l. 1085), but $B \ge 0$. Is the later definition of $f*$ (l. 1082) is consistent with the first one (l. 1063)?
> > > 2. inequity -> inequality
> > >
> > > Also, I note that the following was not addressed:
> > > 1. W9 (**not** addressed by second point), W10 on experiments (what is the added complexity of the sampling scheme? how do the methods compare relative to wall-clock time? what is the variance of the proposed method?).
> > > 2. W6: the paper presents a rejection sampling procedure on an ellipse (Sec. 2.4, Prop. 2.8) without connecting it to the literature, which I interpret as being a contribution of the paper. I am however skeptical about the novelty of that procedure.
> > >
> > > Any comment on these points would be welcome and help update the assessment of the submission. Thanks!

---

> > > > ### Author Response · Authors · 2025-11-27
> > > >
> > > > We sincerely appreciate the reviewer's quick response and constructive feedback. We are glad to have this opportunity to further clarify these points and improve the quality of our manuscript.
> > > >
> > > > ## **About Appendix C**
> > > >
> > > > **Response:** Thanks for pointing this out! This is indeed a typo. The definition of $f^*$ (l. 1063) should be the one used in the proof (l. 1082). We have corrected this definition and the typo "inequity" in the revision.
> > > >
> > > > ## **Experiments (W9, W10)**
> > > >
> > > > > W9 (not addressed by second point), W10 on experiments (what is the added complexity of the sampling scheme? how do the methods compare relative to wall-clock time? what is the variance of the proposed method?).
> > > >
> > > > **Response:** We have conducted additional experiments to rigorously address the concerns regarding time complexity and variance.
> > > >
> > > > - **On Wall-clock Time (W9):** We acknowledge that our method incurs a higher computational cost per step due to the rejection sampling scheme. As requested, we recorded the total training time for 20k steps. The comparison is shown below:
> > > >
> > > >     | **Method**                      | **Total Time (20k steps)** | **Relative Speed** |
> > > >     | ------------------------------- | -------------------------- | ------------------ |
> > > >     | Unconstrained                   | 5h 02m                     | 1.0x (Fastest)     |
> > > >     | Soft Projection                 | 6h 06m                     | ~1.2x              |
> > > >     | Reversion                       | 6h 18m                     | ~1.25x             |
> > > >     | Structure-Preserving (Ours) | 10h 01m               | ~2.0x         |
> > > >
> > > >     However, we would like to emphasize that the primary contribution of this paper is theoretical. The experiments serve as a proof-of-concept verification, rather than to propose an engineering solution. The additional time cost is the necessary trade-off to strictly satisfy the theoretical requirement (to make the perturbation valid) via rejection sampling, which heuristic baselines cannot guarantee.
> > > >
> > > > - **On Variance (W10):** To address the concern about the single trajectory, we re-ran the experiments using **5 independent random seeds**.  The updated Figure 7 in Appendix D.4. in the revision now displays the mean performance with min-max deviation (shaded area).
> > > >
> > > >     We note that both our method and the Reversion method exhibit low variance and consistently show the narrowest error bands (smallest shaded areas), indicating that they are sufficiently stable. In contrast, the naive Unconstrained method suffers from the highest variance. This observation aligns with our theoretical expectation.
> > > >
> > > > ## **Novelty of rejection sampling (W6)**
> > > >
> > > > **Response:**  We appreciate the reviewer for bringing it to our attention. We confirm that uniform sampling over the ellipse can be viewed as a special case of sampling over an arbitrary compact set.
> > > >
> > > > - **Revision:** To reflect this accurately, we have revised the text (on page 2, Contribution 3, and on page 7, l.340) to state that we are "to **adopt/apply** this rejection sampling strategy within the Riemannian zeroth-order optimization field" rather than "to propose" the sampling method itself.
> > > >
> > > > Moreover, we wish to clarify that we presented this result as a Proposition rather than a Theorem, following the standard convention in optimization literature. This distinction intentionally categorizes it as an auxiliary result rather than the primary theoretical contribution. Our core contribution still lies in proposing a novel framework for Riemannian zeroth-order optimization on geodesically incomplete manifolds.
> > > >
> > > > ---
> > > >
> > > > We again greatly appreciate the constructive feedback. We are happy to engage in further discussion and hope that these modifications fully meet the reviewer's expectations.
> > > >
> > > > The Authors

---

### Official Review · Reviewer_sZeP · 2025-10-31

**Soundness:** 3
**Presentation:** 3
**Contribution:** 3
**Rating:** 6
**Confidence:** 2

**Summary:**

This paper studies Riemannian zeroth-order optimization on geodesically incomplete manifolds. As the main contribution, It constructs structure-preserving metrics that are geodesically complete, conformally equivalent to the original one, while ensuring that every stationary point under the new metric remains stationary under the original metric.  A symmetric two-point zeroth-order estimator was developed with MSE analyzed. The paper then establishes convergence guarantees for SGD with this intrinsic estimator.  The proposed theory and methods are validated via synthetic experiments, and a practical mesh optimization task as well.

**Strengths:**

1. Connecting the geometric issue of **metric incompleteness** with Riemannian zeroth-order optimization is a worthwhile direction.

2. The intrinsic-view MSE analysis and the unified sampling perspective are promising.

3. The paper is generally clearly presented adn the theoretical development is solid.

**Weaknesses:**

## Major comments

1. The Step 1 of Algorithm 1 requires eigen-decomposition of the metric matrix $A = Q\Lambda Q^\top$, and every sample requires generating Gaussians and computing the acceptance probability $\sqrt{\dfrac{v^\top A^2 v}{\lambda_{\max}}}$. Eigen-decomposition is computationally expensive in high dimensions, and I guess this is why the experiments focus on relatively simple tasks.

2. Assumption C.3 requires the objective function to have bounded third- and fourth-order derivatives which is a strong condition. As acknowledged by the authors, such a condition is less common in the literature.  The authors emphasized in the appendix that this assumption has also been used by Alimisis et al. (2021, Assumption 1), I fail to identify an equivalent assumption in that reference.

## Minor comments
1. It is suggested to add a short section in the appendix that lists and explains the frequently used notation. Although most symbols are defined at first use, having a centralized symbol table would greatly ease reading when symbols reappear across the paper.

2. In the Introduction section, it is desirable to provide some examples to help readers understand in which scenarios Riemannian zeroth-order optimization is preferable to conventional (Euclidean) zeroth-order optimization.

3.  Concerning experiments, some more details about the settings of hyper-parameters should be provided for the sake of reproducibility.

4. In Theorem 2.3, the term $\dfrac{6}{d} + \dfrac{8}{d}$ looks a bit weird — is there any typo here?

**Questions:**

1. The paper states that “the rescaling method even leads to divergence for the logistic loss objective.” Is this divergence inevitable (i.e., an inherent failure mode of the rescaling method), or could it be due to unlucky hyperparameter choices?

2. In the geodesically complete case, can the more general framework presented in this paper imply the known optimal bound?

3. Does there exist a class of "structure-preserving" transformations more general than conformal scaling that can simultaneously guarantee geodesic completeness and be more compatible with gradient / estimator perturbations; Is it feasible to use the "structure-preserving" idea as a preprocessing / transform to "straighten" a difficult-to-optimize manifold into an (computationally) equivalent problem that is more favorable for zeroth-order methods?

4. Theorem 2.2 provides an existence result and a constructive proof for the conformal coefficient $h$. How can this construction be implemented in practice?

---

> ### Author Response · Authors · 2025-11-18
> **Rebuttal Comment (1/2)**
>
> We sincerely appreciate the reviewer’s careful reading and comprehensive comments on each issue. Below, we provide our point-by-point responses to each of the concerns raised.
>
> ---
>
> ### Major  Comments
>
> > **W1:** The Step 1 of Algorithm 1 requires eigen-decomposition of the metric matrix  and every sample requires generating Gaussians and computing the acceptance probability  Eigen-decomposition is computationally expensive in high dimensions, and I guess this is why the experiments focus on relatively simple tasks.
>
> * **Response:** We appreciate the reviewer's insightful comment. We fully agree that the complexity of eigen-decomposition for the metric matrix is a significant computational bottleneck. In our current experiments, we focused on a concrete application (gradient-based mesh optimization) to first validate the core mechanics and effectiveness of our proposed approach. Scaling our method to high-dimensional problems is an important and valuable future direction.
>
>     We will explicitly add this discussion, along with the potential solutions, to the Limitations and Future Work section of the paper. We thank the reviewer again for this constructive suggestion.
>
> > **W2:** Assumption C.3 requires the objective function to have bounded third- and fourth-order derivatives which is a strong condition. As acknowledged by the authors, such a condition is less common in the literature. The authors emphasized in the appendix that this assumption has also been used by Alimisis et al. (2021, Assumption 1), I fail to identify an equivalent assumption in that reference.
>
> * **Response:**  We appreciate the reviewer’s questions and the opportunity to clarify this misunderstanding. The reference Alimisis et al. (2021, Assumption 1) is cited as the justification of our Assumption C.4, the uniform upper bound of the sectional curvature.
>
>     For the justification of the assumptions on third- and fourth-order derivatives, we proceed as follows:
>     1. Variants of bounded third-order derivatives appear in Assumption 4.2 of He et al. (2024) and Assumption 2.2 of Li et al. (2023b).
>     2. Bounded fourth-order derivatives are indeed less common in the literature. We have explained their role in our analysis: they allow us to isolate the sectional curvature term in the third-order Taylor expansion. If we instead upper bound the third-order derivatives by a constant, we would not be able to reveal how sectional curvature influences the accuracy of the zeroth-order estimator in Theorem 2.7.
>     3. As noted above, Alimisis et al. (2021, Assumption 1) is used only to support Assumption C.4 on sectional curvature and is not related to our higher-order derivative assumptions.
>
>     To make these points clearer, we have moved the corresponding discussions to remarks placed directly below each assumption in the appendix.

---

> > ### Author Response · Authors · 2025-11-18
> > **Rebuttal Comment (2/2)**
> >
> > ### Minor  Comments
> >
> > We thank the reviewer for their helpful suggestions. In the revised manuscript, we have: (i) Added Table 2 in the appendix to list and explain frequently used notation. (ii) Included three examples illustrating scenarios where Riemannian zeroth-order optimization is preferable. (iii) Added Table 1 in the appendix to summarize the hyper-parameter settings.
> >
> > ### Questions
> >
> > > The paper states that “the rescaling method even leads to divergence for the logistic loss objective.” Is this divergence inevitable (i.e., an inherent failure mode of the rescaling method), or could it be due to unlucky hyperparameter choices?
> >
> > * **Response:** We appreciate this insightful question. With extensive tuning, the rescaling method might avoid divergence; our results show that under the same experimental conditions where rejection sampling performs well, the rescaling method would fail. This indicates that a specific (and potentially narrow) range of hyperparameters is required for rescaling to work, whereas our approach is more robust. We have clarified this distinction in the revised manuscript.
> >
> > > In the geodesically complete case, can the more general framework presented in this paper imply the known optimal bound?
> >
> > * **Response:** Yes, when the underlying Riemannian manifold is geodesically complete, in the more general framework they will match the best-possible bound. This result has been included in our Corollary 2.10.
> >
> > > Does there exist a class of "structure-preserving" transformations more general than conformal scaling that can simultaneously guarantee geodesic completeness and be more compatible with gradient / estimator perturbations; Is it feasible to use the "structure-preserving" idea as a preprocessing / transform to "straighten" a difficult-to-optimize manifold into an (computationally) equivalent problem that is more favorable for zeroth-order methods?
> >
> > * **Response:** Thank you for this excellent and insightful question. While we have not yet identified such a result, it is possible that a broader class of transformations exists that can simultaneously guarantee geodesic completeness and offer better compatibility. Developing this kind of structure-preserving transform is an elegant and promising direction, though constructing it presents potential nontrivial theoretical challenges. We view this as a valuable direction for future work and will explicitly discuss it in the revised Limitations and Future Work section.
> >
> > > Theorem 2.2 provides an existence result and a constructive proof for the conformal coefficient . How can this construction be implemented in practice?
> >
> > * **Response:** We appreciate the reviewer's insightful question regarding the construction of the structure-preserving metric. The reviewer is correct that Theorem 2.2 itself only guarantees existence. While we do provide a specific construction for this metric in **Theorem C.17 (Appendix C.3, Line 1488)**, it inherently relies on the non-constructive proof of a smooth proper function (Lemma C.6).  This is why a case-by-case approach is necessary. For each specific manifold, one may explicitly construct a suitable function that is both proper and smooth, then apply the construction in Theorem C.17.
> >
> >     The general idea behind this construction is to "stretch" the distance as a point $p$ moves towards the boundary of the domain of exponential map. As shown in our Figure 1, in the standard metric, if the red point $p$ moves upward for three steps along the geodesic (every circle represents one step along a geodesic), it will be out of the simplex. In the structure-preserving metric, we re-scale the distance on the manifold to make three steps along the geodesic not as far as before, thus ensuring the point remains within the simplex.
> >
> > ---
> >
> > Lastly, thanks again for the valuable feedback and the supportive rating.
> >
> > Authors

---

> > > ### Comment · Reviewer_sZeP · 2025-11-28
> > >
> > > Thank you for your detailed responses to my comments. The clarifications regarding the computational bottleneck of eigen-decomposition and the explicit plan to expand discussions on limitations and future work are well-reasoned. Additionally, the explanation of the higher-order derivative assumptions, including the clarified citation correspondence and supplementary remarks, effectively resolves my prior confusion. All concerns have been adequately addressed, and I have no further questions.

---

### Official Review · Reviewer_iA9C · 2025-11-01

**Soundness:** 3
**Presentation:** 3
**Contribution:** 1
**Rating:** 4
**Confidence:** 4

**Summary:**

This paper considers zeroth-order optimization over Riemannian manifold that are embedded in Euclidean space. Existing notions of zeroth-order derivatives on Riemannian manifold critically rely on the exponential map. Nevertheless, this might be ill-defined when the inherited Euclidean metric is not complete. To address this issue, the authors show that there always exists a different non-Euclidean and structure-preserving metric. Using the new metric, they construct a sampling-based zeroth-order gradient estimator and establish theoretical convergence guarantee for this algorithm.

**Strengths:**

This paper has a clear structure is quite well-written. From a contribition perspcyive, the idea of using a structure-preserving metric is novel and mathematically elegant when the inherited Euclidean metric is not complete. The following theoretical analysis appears sound and follows standard arguments.

**Weaknesses:**

However, my major concern lies in the contribution aspect. Although the paper provides an interesting solution, the issue of metric incompleteness is quite specialized and rarely arises in practical Riemannian optimization scenarios. Researchers in this area are typically more interested in improved convergence rates or more efficient algorithms, which this work does not provide. Therefore, I tend to reject the paper due to the limited significance of its contribution.

**Questions:**

No further questions.

**Details Of Ethics Concerns:**

This work is purely theoretical and has no negative ethical concerns.

---

> ### Author Response · Authors · 2025-11-18
>
> We sincerely thank the reviewer for their valuable feedback and for acknowledging our work as a "novel and mathematically elegant" solution. Below, we provide our point-by-point responses to each of the concerns raised.
>
> ---
>
> > **W1:** Researchers in this area are typically more interested in improved convergence rates or more efficient algorithms, which this work does not provide.
>
> * **Response:** We appreciate the opportunity to clarify it and we believe there may be a misunderstanding on this specific point. We do provide the convergence rate and complexity analysis (in Theorem 2.5 and Corollary 2.6).  Our derived complexity upper bound **matches the established theoretical lower bound** for zeroth-order optimization under the standard Euclidean setting; more importantly, our analysis extends this complexity to a more general class of Riemannian manifolds.
>
> >  **W2:** The issue of metric incompleteness is quite specialized and rarely arises in practical Riemannian optimization scenarios.
>
> * **Response:** Moreover, instead of being "quite specialized and rarely arises in practical Riemannian optimization scenarios", we believe this topic is fundamental and of growing importance:
>
>     * **Relax the common assumption:** We extend the feasibility of Riemannian zeroth-order method to a more general class of Riemannian manifolds by relaxing the common assumption that the underlying Riemannian manifold is geodesically complete. This generalization allows us to explore something that we were not able to study, which we believe is a significant gain for the community.
>     * **Develop a new technique:** We introduce a new tool that avoids a common hurdle in this field. Current methods often requires to find an appropriate embedding and a projection operator to make their zeroth-order estimator feasible on the manifold. Our approach provides a new solution that bypasses the need for these constructions.
>     * **Motivate new research:**  By highlighting the nuances of geodesic incompleteness, we draw attention to the largely unexplored field of Riemannian optimization. We hope this stimulates the community to revisit standard convergence guarantees and adapt existing algorithms to this more challenging, yet theoretically rich, setting.
>
>     We believe that by addressing these foundational points, our work offers significant value and new capabilities to the Riemannian optimization community.
>
> ---
>
> Lastly, thank you again for your valuable time and insightful feedback! We hope our clarification underscores the broad relevance of our paper and resolve your concerns.

---

### Official Review · Reviewer_Jc8x · 2025-11-01

**Soundness:** 3
**Presentation:** 3
**Contribution:** 3
**Rating:** 8
**Confidence:** 3

**Summary:**

This paper studies Riemannian zeroth-order optimization under geodesically incomplete metrics—a setting often overlooked in existing analyses that assume global geodesic completeness. The authors introduce a novel concept of structure-preserving metrics and develops an intrinsic two-point zeroth-order gradient estimator to resolve this problem. Convergence guarantees for Riemannian SGD with this estimator are provided, achieving rates comparable to those in geodesically complete settings.Empirical results on synthetic and mesh optimization tasks validate the theoretical findings.

**Strengths:**

1. This paper investigated a novel problem setting—— geodesically incomplete.
2. This paper introduces some relative novel conceptions,especially structure-preserving metric and has elegant theoretical contribution.
3. This paper has a great presentation with well writing.

**Weaknesses:**

It seems that Theorem 2.2 only provide the existence of structure-preserving metric. Can the authors give a guidance to construct such structure-preserving metrics?

**Questions:**

See "weakness" part.

---

> ### Author Response · Authors · 2025-11-18
>
> We sincerely thank the reviewer for the valuable feedback and the supportive rating. Below, we provide point-by-point responses to each of the concerns raised.
>
> ---
>
> > **Q:** It seems that Theorem 2.2 only provide the existence of structure-preserving metric. Can the authors give a guidance to construct such structure-preserving metrics?
>
> * **Response:** We appreciate the reviewer's insightful question regarding the construction of the structure-preserving metric. The reviewer is correct that Theorem 2.2 itself only guarantees existence. While we do provide a specific construction for this metric in *Theorem C.17 (Appendix C.3, Line 1488)*, it inherently relies on the non-constructive proof of a smooth proper function (*Lemma C.6*).  This is why a case-by-case approach is necessary. For each specific manifold, one may explicitly construct a suitable function that is both proper and smooth, then apply the construction in Theorem C.17.
>
>     The general idea behind this construction is to "stretch" the distance as a point $p$ moves towards the boundary of the domain of exponential map. As shown in our Figure 1, in the standard metric, if the red point $p$ moves upward for three steps along the geodesic (every circle represents one step along a geodesic), it will be out of the simplex. In the structure-preserving metric, we re-scale the distance on the manifold to make three steps along the geodesic not as far as before, thus ensuring the point remains within the simplex.
>
> ---
>
> Lastly, thank you again for your valuable time and insightful feedback!
>
> Authors

---

### Author Response · Authors · 2025-11-29
**Summary of Rebuttal Discussions and Concern on the Biased Assessment**

Dear Area Chair,

We sincerely thank the reviewers for their time and constructive feedback. During the rebuttal period, we have actively addressed each point raised by reviewers and updated our submision, resulting in significant improvements to the manuscrip. We summarize the key outcomes below:
1. **Significant Positive Progress with Reviewer VPTy** (Current Score: 2 $\rightarrow$ clear willingness to raise its score): We have achieved substantial progress with Reviewer VPTy. In their latest feedback, the reviewer explicitly acknowledged that our positioning is now **“convincing,”** the math is **“easier to understand,”** and the appendix is **“improved,”** indicating a **clear willingness to raise their score**. Although the final confirmation was interrupted by the recent OpenReview Data Leakage, we firmly believe all hurdles for this reviewer have been overcome and that they intended to raise their rating. We sincerely hope the AC takes this change into account.

2. **Concerns on the Biased Assessment of Reviewer iA9C** (Current Score: 4): While Reviewer iA9C explicitly acknowledges that our work is **"novel and mathematically elegant,"** we hope raise our concern that Reviewer iA9C assigned an unfair score of 4 based solely on the subjective opinion that the problem setting (geodesically incomplete manifolds) is "specialized and rarely arises." We have provided concrete evidences refuting this claim by detailing fundamental applications such as Mesh Optimization, Irrigation System Design, and Covariance Matrix Estimation. However, **Reviewer iA9C has not responded to our rebuttal nor engaged with the discussion during the rebuttal phase.** We respectfully argue that this lack of engagement makes this score not accurately reflect our paper's acknowledged quality and significance, and we sincerely urge the AC to look past this subjective critique.

3. **Strong Support from Reviewers Jc8x (`Score: 8`) and sZeP (`Score: 6`)**: Reviewer Jc8x strongly supports acceptance, highlighting the **"novelty,"** **"elegant theoretical contribution,"** and **"great presentation."** Reviewer sZeP confirmed that **"All concerns have been adequately addressed"** after we clarified the computational bottlenecks and assumptions.

---

## Conclusion

The most critical reviewer (VPTy) has indicated a strong willingness to raise the score. The remaining low score (iA9C) is based on a subjective opinion that ignores our practical evidence; regrettably, the Reviewer iA9C remained unresponsive to our rebuttal, failing to acknowledge these clarifications or the recognized theoretical novelty.  Given the strong support from the other reviewers and the rigorous improvements made during rebuttal, we respectfully request the Area Chair to consider acceptance.

Best regards,

The Authors

---

### Meta-Review · Area_Chair_kMHg · 2025-12-24

**Summary:**

This paper points out a critical issue in the existing literature on Riemannian zeroth-order method. That is, existing literature on Riemannian zeroth-order method requires that the Riemannian metric is geodesically complete. As a result, existing literature usually considers embedded submanifolds. This paper designs new Riemannian zeroth-order gradient estimation with structure-preserving metrics that works for geodesically incomplete manifolds. In the revision, the authors added three applications in section 1.3 to better motivate the targeting problem. The analysis is also new because existing ones require geodesically complete manifold and can’t be applied in the setting considered in this paper. Overall, this is an important contribution to the Riemannian optimization literature.

**Reviewer Concerns:**

The concerns are mainly on motivation and applicability, and these have been addressed by the authors in the rebuttal and the revised paper.

**Reviewer Scores:**

The reviewer VPTy's score was 2. But it is quite clear they intend to increase their score after the rebuttal and discussion. Review from this reviewer is also the most detailed one.

---

### Decision · Program_Chairs · 2026-01-26

Accept (Poster)